# Functional network collapse in neurodegenerative disease

Jesse A. Brown [1,2] ✉, Alex J. Lee[1], Kristen Fernhoff[1], Taylor Pistone[1], Lorenzo Pasquini[1], Amy B. Wise[1], Adam M. Staffaroni [1], Maria Luisa Mandelli[1], Suzee E. Lee[1], Adam L. Boxer [1], Katherine P. Rankin [1], Gil D. Rabinovici [1], Maria Luisa Gorno Tempini[1], Howard J. Rosen[1], Joel H. Kramer[1], Bruce L. Miller[1] & William W. Seeley [1] ✉

Cognitive and behavioral deficits in Alzheimer's disease (AD) and fronto-temporal dementia (FTD) arise alongside gray matter atrophy and altered functional connectivity, yet the structure-function relationship across the dementia spectrum remains unclear. Here we combine structural and functional MRI from 221 patients—AD ($n = 82$), behavioral variant FTD ($n = 41$), corticobasal syndrome ($n = 27$), and nonfluent ($n = 34$) or semantic ($n = 37$) variant primary progressive aphasia—and 100 cognitively normal individuals. Partial least-squares regression reveals three structure–function components. Component 1 links cumulative atrophy to sensorimotor hypo-connectivity and hyper-connectivity in association cortical and subcortical brain regions. Components 2 and 3 tie focal, syndrome-specific atrophy to peri-lesional hypo-connectivity and distal hyper-connectivity. Structural and functional component scores explain 34% of the variance in global and domain-specific cognitive deficits on average. The functional connectivity changes reflect alterations of intrinsic activity gradients. Eigenmode analysis shows that atrophy relates to reduced gradient amplitudes and narrowed phase angles between gradients, offering a mechanistic account of network collapse in neurodegeneration.

Cognitive deficits in Alzheimer-type dementia (AD) and fronto-temporal dementia (FTD) are associated with tissue degeneration in specific brain regions[1–11]. Brain functional connectivity alterations are also common, involving both decreases and increases in connectivity when compared to cognitively unimpaired individuals[12–18]. A major challenge in clinical neuroscience is to understand the relationship between structural and functional alterations and what overlapping or unique contributions they make to cognitive impairment[19–23]. Progress on this question requires datasets and methods that can unravel the subtypes and stages of atrophy[24] and map these to distinct patterns of functional hypo and hyper-connectivity[25].

Recent advances in functional brain activity modeling may help explain structure-function relationships in neurodegenerative disease.

Canonical functional networks are embedded in a small number of spatial gradients that can be identified with dimensionality reduction techniques[26,27]. These gradients appear to represent intrinsic systems that comprise a low-dimensional basis for different activity and connectivity states[28–30]. In the current study, we hypothesized that atrophy-associated functional connectivity alterations could be parsimoniously explained by disruptions of specific spatial gradients. This endeavor required reconciling two key sets of findings. The first set of findings is that patients with AD and behavioral variant FTD (bvFTD) exhibit opposing spatial patterns of hypo and hyper-connectivity[14,31]. AD involves posterior-predominant atrophy and functional connectivity reductions in the default mode network. By contrast, bvFTD involves fronto-insular atrophy and connectivity reductions in the

[1]University of California, San Francisco, Memory and Aging Center, Department of Neurology, Weill Institute for Neurosciences, San Francisco, CA, USA. [2]Radiata Inc, Mill Valley, CA, USA. ✉e-mail: jbrown81@gmail.com; Bill.Seeley@ucsf.edu

salience network[12,32]. The second key set of findings is that patients with Parkinson's disease or AD have weaker functional connectivity in primary sensory networks – regions that are remote from the primary sites of pathology and neurodegeneration – and stronger connectivity in subcortical and/or association networks[33,34]. Thus, at least two different types of atrophy-associated connectivity alteration are apparent: (1) hypo-connectivity near the lesion and hyper-connectivity remote from it; (2) a convergent pattern of sensory network hypo-connectivity and association network hyper-connectivity across syndromes with different atrophy patterns. Intriguingly, anti-correlated networks are unified as opposing poles of individual gradients[30,35], raising the possibility that specific neuropathology patterns and their associated atrophy may be linked to distinct gradients disruptions and hypo /hyper-connectivity as two sides of the same coin.

Here we studied structure-function relationships using a rich dataset of structural and functional MRI scans from 221 patients with Alzheimer's-type dementia, behavioral variant FTD, corticobasal syndrome (CBS), nonfluent and semantic variants of primary progressive aphasia (nfvPPA/svPPA), and 100 age-matched cognitively normal (CN) controls subjects. We identified three principal structure-function components linking different atrophy patterns to specific brain-wide functional connectivity alterations. These structural and functional components made independent contributions to cognitive deficits. Our analysis revealed that functional connectivity decreases and increases were linked to alterations in a small set of intrinsic functional gradients. Specifically, we found evidence that atrophy associates with reductions in gradient amplitude and changes in between-gradient phase coupling. These two processes reflect both the common and distinct patterns of connectivity alteration across patients with different atrophy subtypes and stages.

## Results

### Functional connectivity decreases and increases across the atrophy spectrum

We assessed structure-function relationships in 221 patients across the FTD-AD spectrum and 100 age, sex, scanner, and fMRI head motion-matched cognitively normal controls, comprising the study's primary cohort (Table 1). We measured gray matter atrophy in 246 cortical and subcortical regions using regional W-scores (**Methods**). Task-free functional connectivity (FC) was measured between region pairs. We first aimed to demonstrate that these clinical syndromes collectively involved most of the brain and were well-suited for examining brain-wide structure-function relationships. We flagged a region as 'affected' within a syndrome when five or more patients in that group showed an atrophy W-score greater than 1.5 (Supplementary Fig. 1). Each syndrome affected an average of 85 regions, and collectively across the five syndromes, 217 out of 246 regions were affected. The mean between-syndrome Jaccard index was 0.28, illustrating significant diversity in the spatial atrophy patterns. We used partial least squares regression (PLSR) to identify the primary structure-function

components. The first three structure components had high reproducibility (**Methods** and Supplementary Fig. 2) and were made the focus of this study.

The first structure-function component (SF1) accounted for 51.2% of the brain atrophy variance and captured the relationship between overall mean atrophy and a distributed pattern of FC decreases and increases. Subjects received structure scores (S1–S3) and function scores (F1–F3) for each structure–function (SF) component. This component's structure scores closely tracked overall mean atrophy (r = 0.994). The SF1 structure-function score correlation was r = 0.56, FDR $p < 0.05$ (Fig. 1A, left). The FC pattern associated included more negative FC between primary visual, somatomotor, and auditory regions (Fig. 1B, left). There was more positive subcortical-cortical FC, most strongly between the striatum, thalamus, and motor regions, along with more subtly increased fronto-parietal association cortex connectivity to widespread cortical and subcortical regions. This pattern captured statistically significant differences in FC edge strength between subjects with low/medium/high scores on structure component 1 (Supplementary Fig. 3).

The second and third components captured syndrome-specific atrophy patterns that explained 9.1% and 6.5% of the atrophy variance. The structure-function correlation on Component 2 was $r = 0.49$, FDR $p < 0.05$; (Fig. 1A, **middle**). Patients on the positive end of the Component 2 spectrum had svPPA-like atrophy in the left anterior temporal lobe. This accompanied weakened functional connectivity (lower than controls) from anterior and medial temporal regions, both locally and globally (Fig. 1B, **middle**). These patients showed heightened functional connectivity involving the dorsal attention, visual, and fronto-parietal networks. In contrast, patients at the negative end of the Component 2 spectrum had AD or CBS diagnoses and showed the opposite pattern of atrophy and functional connectivity: atrophy in the right dorsal parietal cortex, sensory-motor cortex, and thalamic areas; peri-atrophy connectivity deficits; and elevated FC in the left anterior temporal lobe. Cognitively normal control subjects appeared in the middle of the spectrum, with minimal brain atrophy and balanced FC in anterior temporal and dorsal parietal-anchored networks. The Component 3 structure-function correlation was $r = 0.68$, FDR $p < 0.05$ (Fig. 1A, **right**). Supplementary Table 1 shows each syndrome's mean overall atrophy, FC, framewise displacement, and scores on structure/function components 1–3.

We used the three structural components to assess syndrome differences in atrophy and functional connectivity. First, we identified the subset of patients for each syndrome expressing the "typical" atrophy pattern. AD patients showed atrophy in the posterior temporal and parietal lobe; bvFTD in the insula, rostral/orbital frontal lobe, and anterior cingulate; CBS in the primary sensory/motor cortex and superior frontal lobe; nfvPPA in the inferior frontal lobe, insula, and premotor cortex; and svPPA in the anterior temporal lobe. We embedded the three atrophy components into two dimensions to visualize the AD−FTD atrophy spectrum (Fig. 2). We also tested for the

## Table 1 | Patient demographic and clinical information

| Diagnosis | *n* | Age | Sex (F/M) | Education (yrs.) | CDR | CDR-SB | MMSE | Scanner (Trio/Prisma) |
|---|---|---|---|---|---|---|---|---|
| **AD** | 82 | 66.6 ± 9.7 | 50/32 | 16.0 ± 3.3 | 0.8 ± 0.3 | 5.1 ± 2.4 | 20.9 ± 5.7 | 29/53 |
| **bvFTD** | 41 | 61.4 ± 10.4 | 17/24 | 16.1 ± 2.3 | 1.1 ± 0.5 | 8.2 ± 3.1 | 24.6 ± 3.1 | 19/22 |
| **CBS** | 27 | 66.3 ± 9.5 | 14/13 | 16.5 ± 2.9 | 0.5 ± 0.3 | 4.2 ± 2.4 | 25.8 ± 3.4 | 17/10 |
| **nfvPPA** | 34 | 70.8 ± 6.3 | 23/11 | 16.7 ± 3.5 | 0.4 ± 0.7 | 3.7 ± 3.8 | 25.9 ± 4.5 | 25/9 |
| **svPPA** | 37 | 64.9 ± 6.9 | 19/18 | 17.1 ± 2.7 | 0.7 ± 0.3 | 5.4 ± 2.7 | 25.5 ± 3.3 | 30/7 |
| **CN** | 100 | 66.5 ± 9.1 | 56/44 | 17.6 ± 2.0 | 0.0 ± 0.0 | 0.0 ± 0.0 | 29.2 ± 1.3 | 53/47 |

*AD* Alzheimer's disease, *bvFTD* behavioral variant FTD, *CBS* corticobasal syndrome, nfvPPA nonfluent variant primary progressive aphasia, *svPPA* semantic variant primary progressive aphasia, *CN* cognitively normal, *CDR/CDR-SB* clinical dementia rating/sum of boxes, *MMSE* mini-mental status examination.

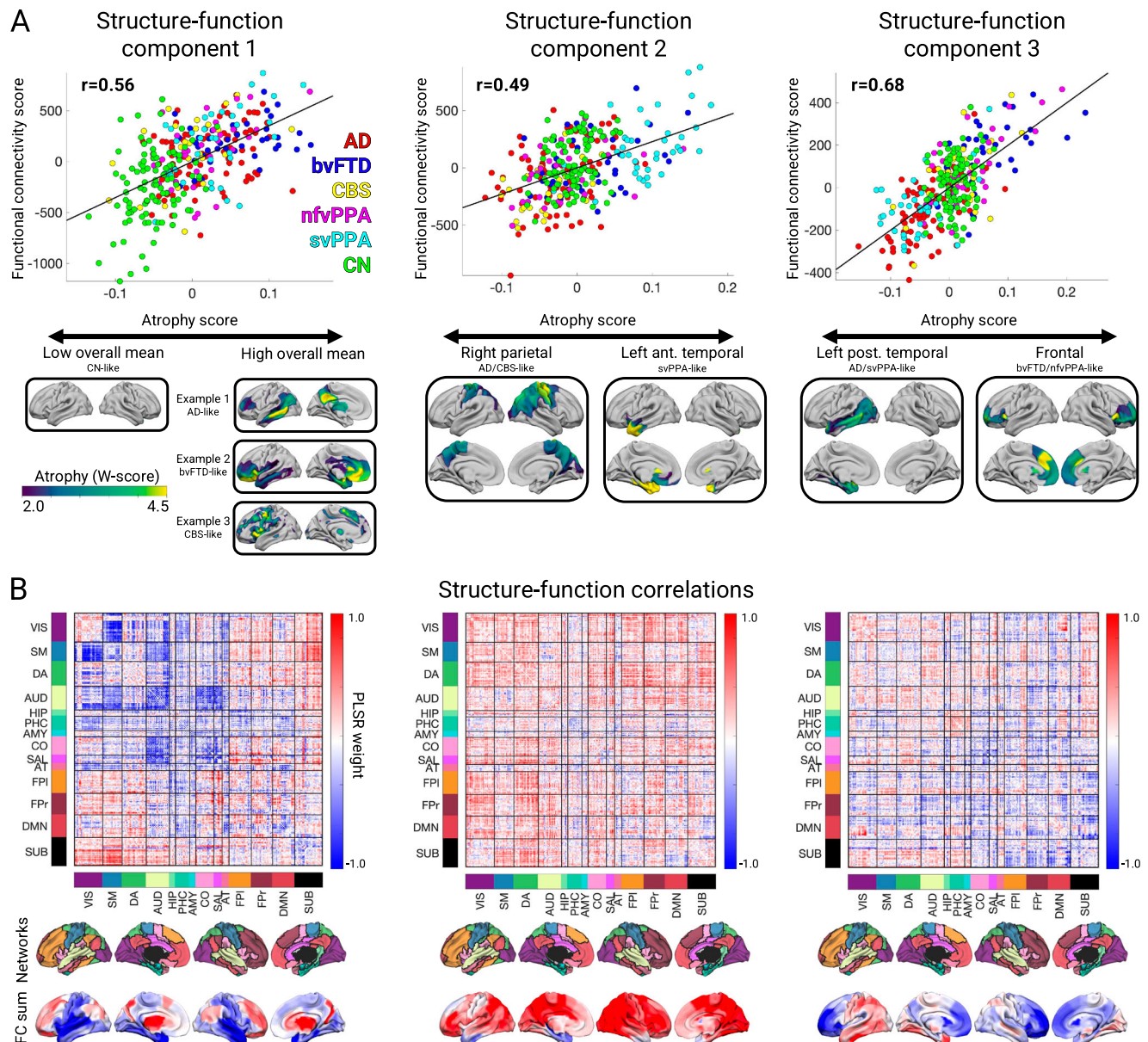

**Fig. 1 | The first three structure-function components across the AD-FTD spectrum. A** The correlation between atrophy and functional connectivity (FC) scores for partial least squares regression components 1-3. Beneath each correlation plot are the associated atrophy patterns associated with a negative or positive atrophy component score. For component 1, three example patients are shown with different atrophy patterns but equivalent overall mean atrophy. **B** Matrices showing the partial least squares FC weights for each component, along with the network membership for each brain region. Negative and positive weights indicate a decrease or an increase in FC with an increase in the atrophy component score (and vice versa). Matrices show 14 brain FC networks for reference. The top row of brain surfaces shows the 14 networks. The bottom row of brain surfaces shows the FC PLSR region sums. VIS visual, SM sensory-motor, DA dorsal attention network, AUD auditory, HIP hippocampal, PHC parahippocampal, AMY amygdala, CO cingulo-opercular, SAL salience, AT anterior temporal, FPl left fronto-parietal, DMN default mode network, FPr right fronto-parietal, SUB subcortical. AD Alzheimer's Disease. bvFTD behavioral variant FTD. CBS corticobasal syndrome, nfvPPA non-fluent variant primary progressive aphasia, svPPA semantic variant primary progressive aphasia, CN cognitively normal.

presence of meaningful atrophy stages and subtypes within each syndrome. We found evidence for significant atrophy stage variation within each syndrome, and for the presence of two likely atrophy subtypes in AD, bvFTD, CBS, and svPPA (**Supplementary Results** and Supplementary Fig. 4).

Next, we assessed the functional connectivity alterations associated with the typical atrophy patient subset for each syndrome. The associated functional connectivity alterations are shown in Fig. 2. We tested how well those FC patterns could be explained in terms of the three functional components. For each syndrome, the best match to the true FC difference matrix (Fig. 2, upper triangles) was the

corresponding reconstructed FC matrix (Fig. 2, lower triangles) (AD, actual versus reconstructed $r = 0.81$; bvFTD $r = 0.82$; CBS $r = 0.41$; nfvPPA $r = 0.50$; svPPA $r = 0.82$, all FDR $p < 0.05$). This confirmed that three structure-function components captured the predominant syndrome-associated patterns.

We confirmed the structure-function relationship reliability in two ways. Using ridge regression with four-fold cross-validation, each of the three structure–function components had significant out-of-sample correlation (median $r = 0.32$–$0.49$; all FDR $p < 0.05$; Supplementary Fig. 5). The same associations replicated in an independent multi-site ADNI cohort ($n = 477$ across 35 sites; $r = 0.08$–$0.25$; all

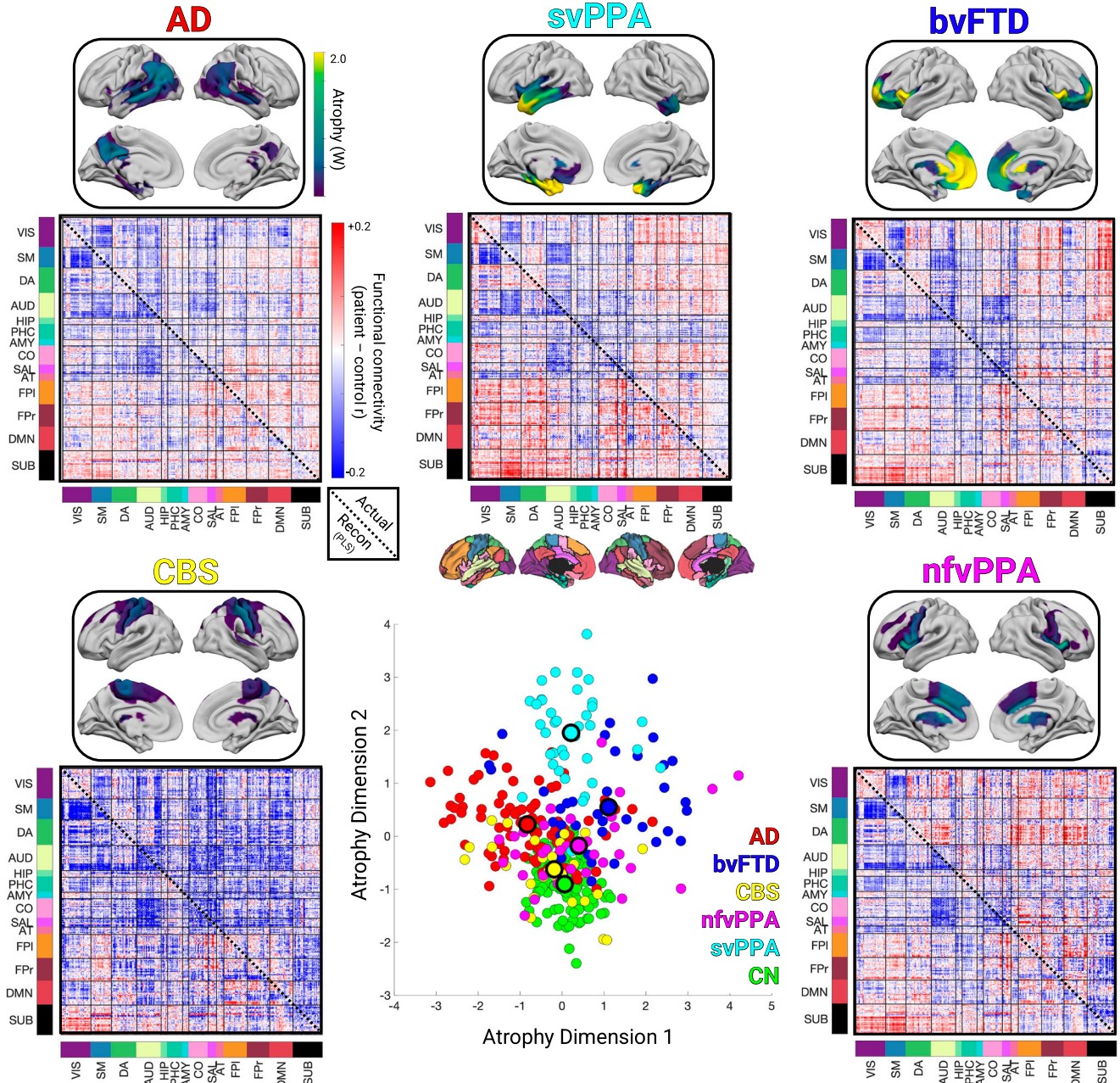

**Fig. 2 | Atrophy and functional connectivity patterns for each syndrome.**
Central scatter plot depicts individual subjects' atrophy similarity in two dimensions based on multidimensional scaling of structure components 1-3. Large dots show the mean position for each clinical syndrome. For each syndrome, the mean atrophy pattern and functional connectivity matrix are shown for the set of "typical" patients that were accurately classified as having that syndrome based on their individual atrophy pattern. The mean functional connectivity difference matrix is shown for the typical patients for each syndrome versus cognitively normal subjects (upper triangles), along with the reconstructed matrix based on function components 1-3 (lower triangles). AD Alzheimer's Disease, bvFTD behavioral variant FTD, CBS corticobasal syndrome, nfvPPA nonfluent variant primary progressive aphasia, svPPA semantic variant primary progressive aphasia, CN cognitively normal.

FDR $p < 0.05$; Supplementary Fig. 6). Collectively, the three main structure-function components in AD and FTD captured specific atrophy patterns and associated hypo/hyper-FC profiles that replicated in different individuals, syndromes, and MRI scanners.

### Neuropsychological correlations with structural and functional components

We next examined the relationship between the brain structure-function components and cognitive performance. We first focused on two tests of global functioning, the CDR® + NACC-FTLD sum of boxes (Miyagawa et al. [36]; henceforth referred to as CDR-SB) and

MMSE. A generalized additive model was used to estimate cognitive scores based on the first three structure and function scores, either as linear or non-linear terms, and covariates (**Methods**). The model for CDR-SB explained 50% of the variance. The strongest predictors were S1 ($F = 22.88$, FDR $p < 0.05$; Fig. 3A/B), F1 ($F = 17.16$, FDR $p < 0.05$), and S3 ($F = 10.74$, FDR $p < 0.05$). This indicated that patients with the most severe clinical impairment had high overall mean atrophy (S1), most pronounced in the frontal lobe (S3), along with subcortical hyper-connectivity and primary sensory cortical hypo-connectivity (F1). For MMSE, the model explained 40% of the variance with the strongest predictions from S1 (F = 22.31, FDR

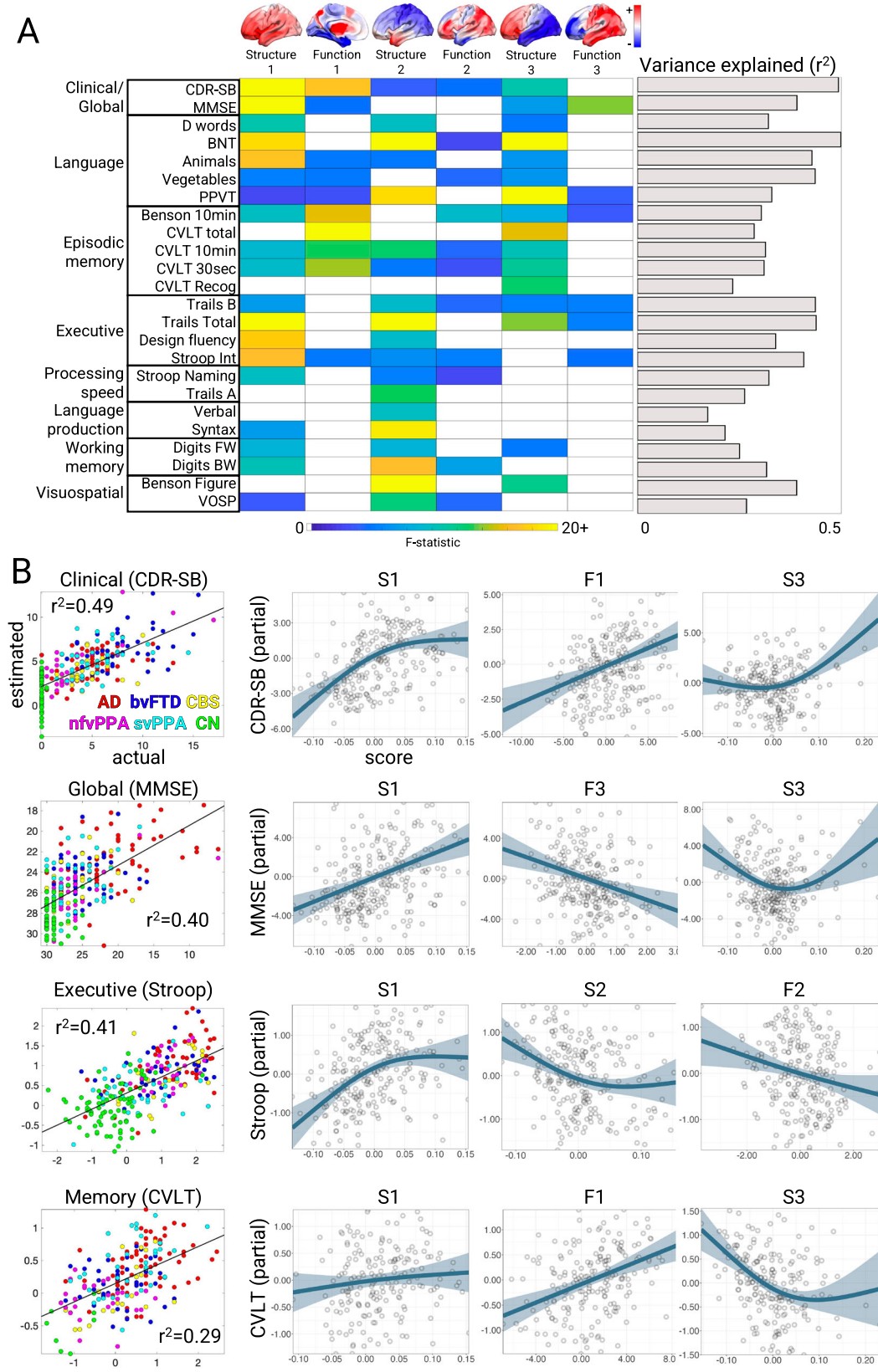

*p* < 0.05), F3 (F = 13.94, FDR *p* < 0.05), and S3 (*F* = 7.22, *p* = 0.001). S3 had significant nonlinearity (Fig. 3B), such that subjects with either frontal or temporal atrophy had equivalently poor MMSE scores. Subjects with worse MMSE scores also had anterior connectivity deficits and posterior connectivity enhancements. Thus, CDR-SB and MMSE scores significantly correlated with the overall amount of atrophy, specific atrophy patterns, and distinct functional connectivity alterations.

We then assessed the relationship between brain structure-function component scores and neuropsychological test scores for episodic memory, working memory, processing speed, executive function, visuospatial processing, speech, and language. Across

**Fig. 3 | Neuropsychological correlates of brain structure-function components. A** Cognitive test score estimates based on a generalized additive model with brain structure-function component scores and covariates. The F-statistic for the structure and function terms in each model are shown when significant ($p < 0.05$, FDR-corrected) or trending ($p < 0.05$, uncorrected); the F-statistic is a positive-only one-sided test. The overall variance explained for each cognitive test is also shown. Brains show the weight of each region for structure components and the sum of region FC edge weights for function components. **B** Correlation between actual and estimated test scores for clinical dementia severity (CDR-SB), global cognition (MMSE), executive functioning (Stroop Interference), and episodic memory (California Verbal Learning Test, CVLT total). Test scores are presented with higher values representing worse performance, and converted to Z-scores for Stroop and CVLT. Partial effect plots are shown for three predictors of interest for each test. Positive relationships indicate a correlation between the positive structure/function pattern and the neuropsychological score. Shaded bands show the mean ± 2x standard errors of the fit (95% confidence interval). BNT Boston Naming Tes, PPVT Peabody Picture Vocabulary Test, VOSP Visual Object and Space Perception, AD Alzheimer's Disease, bvFTD behavioral variant FTD, CBS corticobasal syndrome; nfvPPA: nonfluent variant primary progressive aphasia, svPPA semantic variant primary progressive aphasia, CN cognitively normal.

domains, the models explained an average of 34% of the variance (Fig. 3A; mean $r^2 = 0.34 \pm 0.09$; range=0.17–0.50). The brain-behavior relationships are clustered into two groups. The first group represented dysfunctions in clinical/global functioning, language, and episodic memory, most strongly influenced by S1, F1, and S3. This pattern was most strongly exhibited by patients with AD or svPPA, with high mean atrophy, most prominent in the temporal lobe, along with sensory hypo-connectivity and subcortical/association hyper-connectivity. The second group included dysfunctions in executive function, processing speed, language production, and visuospatial processing, associated most strongly with S1 and S2. The patients most strongly represented in this group had CBS, nfvPPA, or AD, also with high mean atrophy, but focused in the parietal lobe. Overall, brain-behavior relationships were strongest for structural components, though key functional relationships predicted global cognition and memory performance.

We evaluated the prospective validity and reliability of the structure-function-cognition relationship in three analyses (**Supplementary Results**). In the UCSF cohort with longitudinal cognition, higher baseline F1 predicted faster MMSE decline over three years (time × baseline F1: $t = -2.72$, FDR $p < 0.05$; Supplementary Fig. 7). In the independent ADNI cohort ($n = 774$ scans), the same structure–function predictors explained 33% of CDR-SB variance (all FDR $p < 0.05$). In the ADNI longitudinal subset, within-subject changes in F1 and S1 tracked changes in CDR-SB at the trend level (uncorrected $p < 0.05$; Supplementary Fig. 8). Together, these results support cross-cohort generalizability of the structure–function measures and suggest that F1 may have prognostic utility.

### Low-dimensional functional connectivity changes associate with different atrophy components

All three primary structure-function components involved hypo and hyper-FC. We hypothesized that specific alterations in low-dimensional brain activity underlie the hypo/hyper-FC patterns for each functional component. This was tested by performing PCA dimensionality reduction on the fMRI timeseries data to derive spatial components, henceforth referred to as gradients (Fig. 4A), and their associated temporal fluctuations (**Methods**). Specifically, we derived the gradient maps from an independent cohort of age-matched cognitively normal subjects ($n = 321$) and projected all primary cohort patient and control fMRI data into this space. This approach assumed spatial gradient patterns are stable regardless of disease and that atrophy perturbs gradient temporal dynamics. The spatial patterns captured known gradients including unipolar sensory-to-association (Gradient 1), sensory-to-cognitive (Gradient 2), visual-to-sensorimotor (Gradient 3), task-negative-to-task-positive (Gradient 4), and left-right asymmetric (Gradient 6)[26,30]. We confirmed that the gradient spatial maps derived from the independent cohort were highly similar to those obtained from the primary cohort (Supplementary Fig. 9).

We then assessed how functional connectivity component F1-F3 scores related to specific across-subject differences in the gradient temporal variance and covariance. We found that all three function components had strong and specific relationships with the six primary

gradients, explaining most of the across-subject FC statistical variance. For F1, the first six gradients explained 82.0% of the functional connectivity statistical variance, with the biggest contributions from Gradient 1 temporal variance (26.4% of F1 statistical variance, $t = -17.08$; FDR $p < 0.05$ for all reported terms; Fig. 4B/C), Gradient 1/4 covariance (15.1% of F1 statistical variance, $t = +10.32$; Fig. 4B/D), and Gradient 1/5 covariance (10.8% of F1 statistical variance, $t = +10.11$). Subjects with higher overall mean atrophy had lower Gradient 1 variance, which, because of its unipolar nature (all regions positive), reflected lower global BOLD signal amplitude. The correlation of Gradient 1 variance and global signal amplitude was $r = 0.91$ ($p < 0.05$). This explained the weakened global functional connectivity, most extreme in sensory-motor regions with the largest Gradient 1 weights. Subjects with high overall mean atrophy also had stronger Gradient 1-4 covariance than low atrophy subjects. This resulted in more positive functional connectivity between regions with positive weights on Gradient 4 (dorsal attention, cingulo-opercular) and positive weights on Gradient 1 (all regions). In contrast, regions with negative weights on Gradient 4 (default mode) had a negative correlation with Gradient 1 and therefore with the whole brain. Thus, atrophy-associated changes in coupling of Gradient 1 with other gradients were mathematically equivalent to specific networks increasing or decreasing their integration with the rest of the brain.

Function components 2 and 3 also had strong relationships with gradient activity. Gradient activity explained 86.2% of the F2 statistical variance, with the strongest influence by Gradient 1 temporal variance (58.3% of statistical variance, t = +31.02), Gradient 1/2 covariance (10.8% of statistical variance, $t = +10.82$; Fig. 4B/E), and Gradient 1/6 covariance (5.3% of statistical variance, t = -8.93). F3 had 55.2% of its statistical variance explained by Gradient 1/2 covariance (14.0% of variance, $t = -8.20$; Fig. 4B/F), Gradient 2 variance (10.2% of statistical variance, t = -6.97), and Gradient 2/3 covariance (6.8% of statistical variance, $t = -5.21$). Here, subjects with greater frontal atrophy had lower Gradient 2 temporal variance, reflecting reduced within-network FC for regions on either pole of the gradient, and also less anticorrelation (i.e. stronger FC) between regions at opposite gradient poles. Overall, the majority of the atrophy-associated FC statistical variance was explained by six gradients and their interactions, indicating a low-dimensional basis for the observed FC alterations.

### Gradient phase and amplitude changes reflect hypo and hyper-connectivity patterns

The observation that atrophy and FC were associated with gradient activity prompted a question: can we simulate these altered FC patterns with a generative model and if so, what can that model tell us about how atrophy disrupts brain activity dynamics? We modeled the low-dimensional brain activity patterns ("gradients") as a coupled oscillator system, where each gradient behaves like a source that fluctuates over time according to its own dynamics and its interactions with the other sources (**Methods**). We obtained these sources empirically using data-driven dimensionality reduction[37] and have previously shown that such components correspond to spatial activity

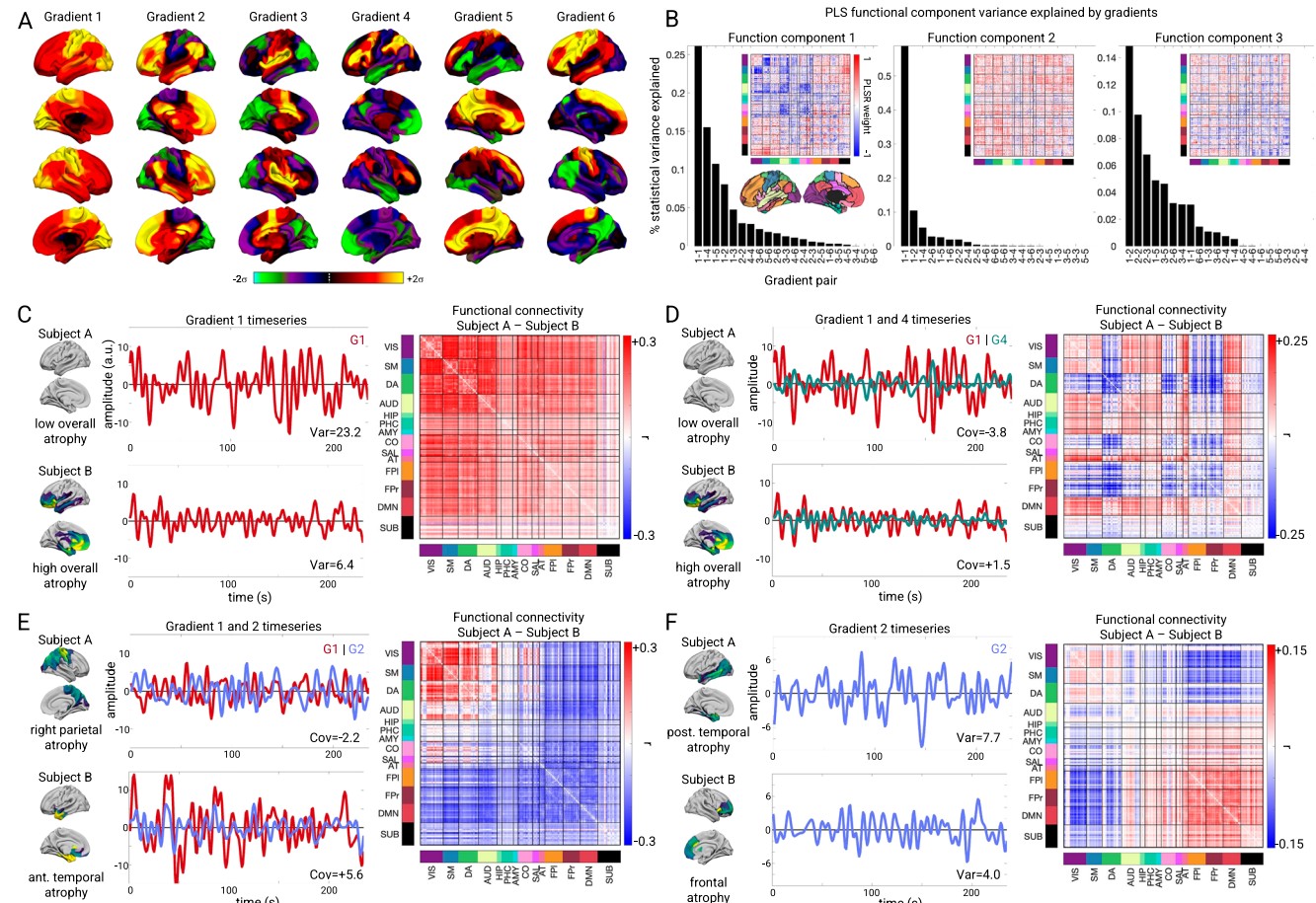

**Fig. 4 | Low-dimensional gradient activity relationship with structure-function components. A** Gradient spatial weight maps based on PCA of fMRI timeseries data from the independent cognitively normal cohort (n = 321). Weights represent PCA loadings. **B** Statistical variance explained in each PLSR functional component by individual gradient temporal variances (e.g., 1-1, 2-2) and gradient pair temporal covariances (e.g. 1-2, 2-3). Insets show PLSR component functional connectivity (FC) edge weight matrices. **C–F** Select gradient timeseries and associated FC differences. **C, D** Gradient 1 temporal variance, Gradient 1-4 covariance, and FC differences for two subjects with low or high overall mean atrophy. **E** Gradient 1-2 covariance and FC differences for two subjects with right parietal or left anterior temporal atrophy. **F** Gradient 2 temporal variance and FC differences for two subjects with posterior temporal or frontal atrophy. VIS visual, SM sensory-motor, DA dorsal attention network, AUD auditory; HIP: hippocampal, PHC para-hippocampal, AMY amygdala, CO cingulo-opercular, SAL salience, AT anterior temporal, FPl left fronto-parietal, DMN default mode network, FPr right fronto-parietal, SUB subcortical.

gradients[30]. From each subject's gradient timeseries, we estimate a linear dynamical system—essentially, how the acceleration of each gradient depends on the current level and velocity of all gradients—which yields a coupling matrix (Fig. 5A)[38]. Taking the eigendecomposition of this matrix produces a set of eigenmodes that summarize the system's characteristic patterns of co-fluctuation. Each eigenmode oscillates at a single frequency and assigns every gradient an amplitude (how strongly it participates in that mode) and a phase angle (its timing relative to the others). The observed brain activity at any moment can be expressed as the sum of these modes. Changes in gradient amplitudes or in the phase angles between gradients therefore, provide a compact way to describe how disease alters large-scale functional coupling. Here, we computed each subject's gradient coupling parameters, derived the eigenmodes, and confirmed that the eigenmodes faithfully reconstructed individual FC fingerprints[39] (Supplementary Fig. 10). This supported the use of eigenmode analysis for understanding FC differences.

As an illustration of how the eigenmodes captured different FC patterns, we computed the eigenmodes for two groups of subjects (n = 5) with the lowest or highest overall mean atrophy as measured by structure-function component 1 (Fig. 5B). We used these group-specific eigenmodes to simulate gradient timeseries and compute FC matrices (Fig. 5C). The simulated FC differences between low and high atrophy subjects revealed the same pattern of atrophy-associated FC changes represented on SF1 – reduced primary sensory FC, elevated subcortical-cortical FC, and elevated fronto-parietal association FC. This indicated that the eigenmodes contained sufficient information to explain the observed hypo/hyper-FC patterns.

This observation suggested a potential three-way relationship between FC, eigenmodes, and gradient temporal variance/covariance. We hypothesized that two eigenmode-derived quantities would relate to gradient temporal variance/covariance, and thus to atrophy-associated FC alterations: 1) the net amplitude of each gradient across all six eigenmodes, which would be related to gradient temporal variance, and 2) the average phase angle between each pair of gradients across all modes, which would be related to the gradient temporal covariance. We statistically evaluated this by measuring the correlation between eigenmode gradient amplitude/angle and gradient temporal variance/covariance. These quantities were strongly related (Supplementary Fig. 11; corresponding: median absolute $r = 0.77$, $p = 6.21 \times 10^{-65}$; non-corresponding: median $r = 0.07$, $p = 0.19$). This demonstrated that atrophy-related shifts in brain-wide FC corresponded to decreasing gradient amplitudes and alterations in the typical phase angle between pairs of gradients.

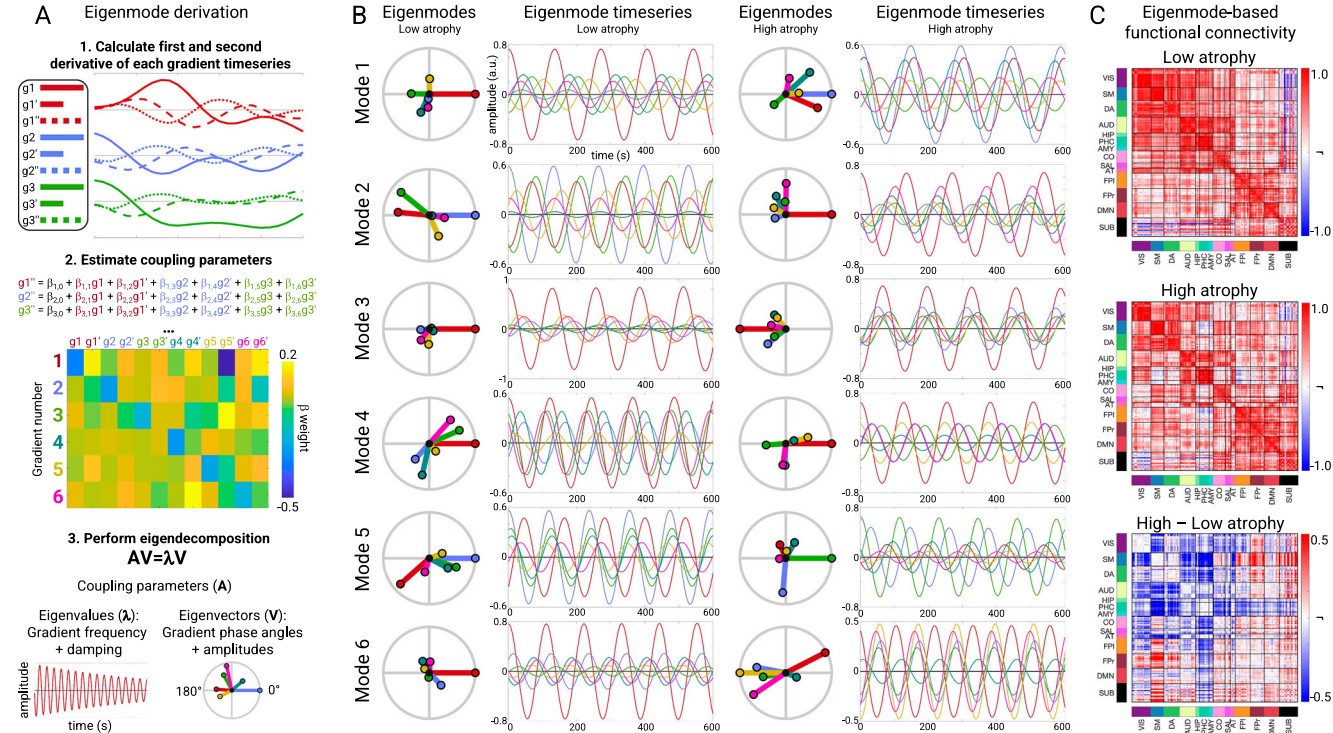

**Fig. 5 | Eigenmode derivation and properties in low and high atrophy subjects.** **A** Procedure for deriving gradient coupling parameters and eigenmodes. **B** The six eigenmodes for two example subgroups (*n* = 5) with the lowest or highest total atrophy. Eigenmodes are ordered from lowest to highest frequency. Circle plots show the phase angle and amplitude of each gradient on each eigenmode. Timeseries plots show the resultant gradient oscillations, occurring at the eigenmode-specific frequency. **C** Eigenmode-based FC matrices for low and high atrophy subjects and the FC difference matrix. VIS visual, SM sensory-motor, DA dorsal attention network, AUD auditory, HIP hippocampal, PHC parahippocampal, AMY amygdala, CO cingulo-opercular, SAL salience, AT anterior temporal, FPl left fronto-parietal, DMN default mode network, FPr right fronto-parietal, SUB subcortical. Portions of **A**. are reprinted from[30] with permission from Elsevier.

We further examined the gradient amplitude and phase angle properties for four gradient relationships most strongly associated with the structure-function components. Subjects with lower Gradient 1 temporal variance had either higher overall mean atrophy (SF1) or right parietal atrophy (SF2), which significantly correlated with Gradient 1 amplitude (*r* = 0.66, FDR *p* < 0.05; Fig. 6A). This reduced amplitude resulted in globally reduced functional connectivity strength because of Gradient 1's unipolarity. Subjects with high overall mean atrophy also had increased Gradient 1–4 temporal covariance, which negatively correlated with Gradient 1–4 phase angle (*r* = −0.80, FDR *p* < 0.05; Fig. 6B). The phase angle progressively decreased from 93° for subjects in the 20th percentile to 64° for subjects in the 80th percentile. This smaller angle reflected more positive temporal correlation between the Gradient 1 and 4 time series, resulting in hyper-connectivity of regions with the same sign on each gradient (+/+ or −/−) and hypo-connectivity of regions with opposite signs (+/−). We interpreted a shift away from 90° as a "collapse", given that the gradients were identified as temporally orthogonal (i.e., uncorrelated) components in cognitively normal control subjects. For Gradients 1 and 2, there was again a negative correlation between temporal covariance and phase angle (*r* = -0.80, FDR *p* < 0.05; Fig. 6C). In this case, the collapse away from 90° was in one of two directions, depending on the atrophy pattern. Subjects with posterior temporal atrophy (SF3) or anterior temporal atrophy (SF2) showed an increase to 109° (10th percentile) while subjects with frontal or right parietal atrophy showed a decrease to 57° (90th percentile). This bi-directional collapse resulted in opposite patterns of hypo and hyper-connectivity, resulting from regions with positive or negative weights on Gradient 1 and 2 coming into phase or going out of phase. Finally, Gradient 2 temporal variance significantly correlated with amplitude (*r* = 0.74, FDR *p* < 0.05; Fig. 6D). Subjects with posterior temporal atrophy (SF3) had greater

Gradient 2 amplitude than those with frontal atrophy. This reflected more extreme correlated fluctuations for regions with the same sign on Gradient 2 (+/+, −/−) and anticorrelated fluctuations for regions with opposite signs (+/−). Overall, eigenmode analysis revealed that each atrophy-associated hypo/hyper-connectivity pattern was linked to specific gradient amplitude alterations, reflected in gradient temporal variance changes, and phase angle alterations, reflected in gradient pair temporal covariance changes.

## Discussion

Here we analyzed structural and task-free functional MRI scans from patients across the AD-FTD spectrum to identify relationships between gray matter atrophy and functional connectivity. We discovered three reproducible structure-function components. The primary component represented a relationship between overall mean atrophy, regardless of spatial location, hypo-connectivity in primary cortical regions, and hyper-connectivity in subcortical and fronto-parietal association cortex regions. The second and third components linked focal syndrome-specific atrophy patterns to peri-lesional hypo-connectivity and distal hyper-connectivity. These structural and functional alterations collectively correlated with impairments in global and domain-specific cognition. Each functional component could be accounted for by variance in six intrinsic activity gradients derived from cognitively normal individuals, suggesting that the disease-related functional alteration patterns are constrained by the brain's intrinsic functional architecture. Eigenmode analysis of the gradient temporal dynamics revealed reduced amplitude of specific gradients and collapsed phase angles between gradients, offering a possible explanation for the observed patterns of hypo and hyper-connectivity.

Three brain atrophy components explained two-thirds of the variance in atrophy across this diverse set of AD and FTD syndromes

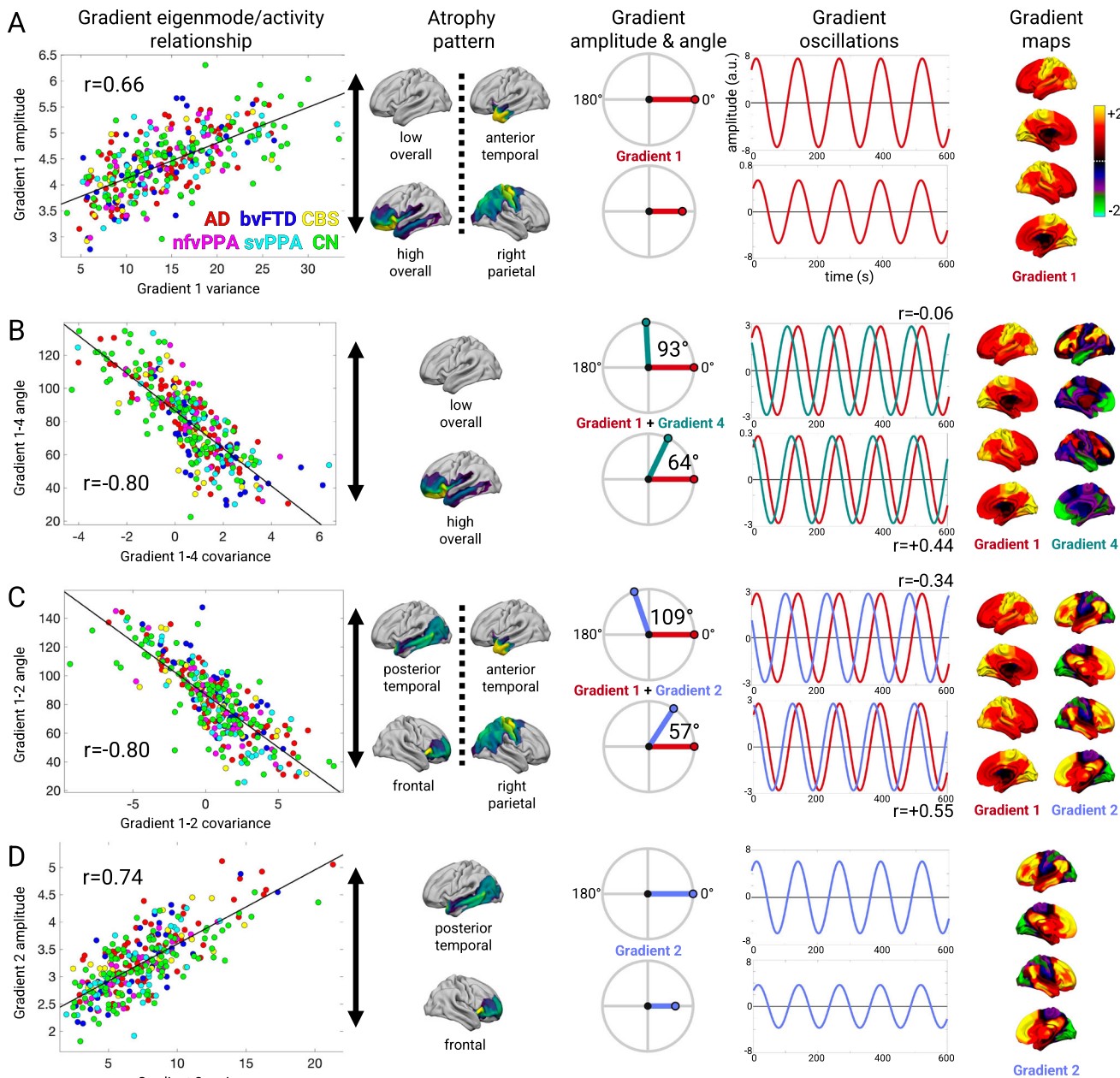

**Fig. 6 | Gradient eigenmode relationships with different structure-function components. A–D** Left to right: the correlation between each subject's gradient temporal variance/covariance and gradient amplitude/angle; the atrophy components associated with higher or lower gradient amplitude/angle; illustration of the gradient amplitude/angle differences and the resultant simulated timeseries; and the gradient spatial maps. **A** Gradient 1 amplitude associated with atrophy component 1. Gradient amplitudes and timeseries are shown for subjects in the 20th and 80th percentile of atrophy component 1. **B** Gradient 1-4 angle associated with atrophy component 1. Gradient phase angles and timeseries are shown for subjects in the 20th and 80th percentile of atrophy component 1. R-values show correlation between the simulated gradient timeseries. **C** Gradient 1-2 angle associated with atrophy components 3 and 2. Gradient phase angles and timeseries are shown for subjects in the 10th and 90th percentile. **D** Gradient 2 amplitude associated with atrophy component 3. Gradient amplitudes and timeseries are shown for subjects in the 20th and 80th percentile of atrophy component 3. AD: Alzheimer's Disease; bvFTD: behavioral variant FTD; CBS: corticobasal syndrome; nfvPPA: nonfluent variant primary progressive aphasia; svPPA: semantic variant primary progressive aphasia; CN: cognitively normal.

that collectively involved nearly all cortical and subcortical regions. The first atrophy component captured the mean overall atrophy. This corresponded to a mean spatial atrophy pattern most pronounced in cingulo-insular regions, consistent with the FTD-predominant composition of this dataset[40]. The second and third components described opposing focal atrophy patterns in the left-predominant temporal pole versus right-predominant dorsal parietal cortex (component 2) and left-predominant temporal-occipital areas versus prefrontal-insula-cingulate areas (component 3). These three components collectively stratified patients with different syndromes, atrophy stages, and

subtypes, consistent with well-characterized atrophy subtypes in different AD and FTD syndromes[5,41–47]. We used partial least squares regression to derive atrophy components and continuous scores so that we could link progressive atrophy patterns across stages to corresponding functional connectivity alterations. It may not always be appropriate to represent patients with AD and FTD along such a continuum, as these syndromes are often caused by distinct neuropathological diseases that involve specific cell populations, subcortical nuclei, or cortical layers[48–51]. Indeed, our choice to only include patients with a high or medium confidence clinical diagnosis, while

intended to create a clean cohort well-suited for structure-function characterization, may have exaggerated the presence of distinct clinical-anatomical subgroups. Future work may assess structure-function relationships in distinct categorical groups, such as those with homogenous underlying pathological substrates.

Distinct functional hypo/hyper-connectivity patterns were associated with each atrophy component. We did not anticipate finding structure-function component 1's convergent pattern of FC alteration in patients with different syndromes and heterogeneous atrophy patterns. However, recent fMRI studies in Parkinson's disease and Alzheimer's disease have reported similar patterns of primary sensory connectivity decrease and subcortical and/or association network connectivity enhancements[33,34,52], regions that are remote from the primary sites of pathology and neurodegeneration. There are several ways connectivity can be altered in regions without structural damage. Classical diaschisis involves a focal lesion causing depressed function in a structurally intact target region[53], while connectomal diaschisis includes network-wide alterations[54]. Here, we assessed widespread FC alterations by deriving summary scores capturing different FC patterns and then linked these scores to underlying low-dimensional gradient activity levels. We found that the principal FC pattern was based on reduced Gradient 1 amplitude and increased Gradient 1-4 coupling, corresponding to a reduced phase angle between these gradients. A critical question is how atrophy in disparate locations can associate with convergent alterations in these functional systems. Previous studies have reported widespread FC decreases and increases following focal stroke lesions[55], with weak spatial correspondence between structural and functional alterations[56]. This suggests that focal structural damage can be associated with non-local disruptions of large-scale network dynamics. Given the evidence that brain activity dynamics occur in a low-dimensional functional state space[20,29,30], it may be expected that disparate lesions will converge on a limited range of FC perturbations. In the healthy brain, the optimal set point for each gradient may be temporal orthogonality, maximizing the spatiotemporal segregation of different networks[57] and the potential entropy of the brain[58,59]. Previous studies have shown that a loss of high-frequency local interactions between brain regions can result in phase collapse, hypersynchrony, and rsfMRI overconnectivity[60]. Here, one possibility is that structural damage may preferentially impair local functional interactions, leading to phase collapse between gradients and resulting in hypo and hyper-connectivity between different networks as two sides of the same coin. This may occur in a convergent or divergent fashion, as we found with SF1 and SF2/3, respectively. The linkage between functional connectivity, activity gradients, and eigenmodes raises a question about whether any of these processes are epiphenomenal. Gradients represent standing waves that combine by superposition to create activity flow patterns described by eigenmodes[61,62]. Future studies may consider how atrophy perturbs activity flow and whether damaged structural connections play a key role.

The structure-function components explained 25–50% of the variance in cognitive deficits in this patient cohort with clinical dementia. These deficits clustered into two groups: clinical/global functioning, language, and episodic memory, most strongly influenced by S1, F1, and S3; and executive function, processing speed, language production, and visuospatial processing, correlated with S1 and S2. This clustering is consistent with the emerging recognition that diverse brain lesions can converge on low-dimensional behavioral deficits[22,63]. The structural components tended to be stronger predictors of cognitive deficits than functional components, which is perhaps unsurprising given that the syndrome diagnostic criteria include structural neuroimaging features. Nonetheless, functional components did explain significant variance in cognitive deficits. Most strikingly, F1 was a significant predictor of CDR-SB over and above the amount of overall mean atrophy (S1 score). F1 may represent a neural substrate for cognitive reserve that can compensate for structural degeneration[64]. Across FTD and AD dementia syndromes, functional connectivity

levels have been shown to compensate for atrophy or molecular pathology to preserve cognitive functioning[21,65,66]. Future work may examine whether subjects with worse-than-expected function for their amount of atrophy, as measured by the residuals from the structure-function regression line, may have more reserve and greater potential response to treatment. Our assessment of structure-function across anatomical and clinical disease stages had several important limitations. Our cohort included individuals with a clinical dementia diagnosis and age-matched cognitively normal controls, excluding preclinical and prodromal stages (subjective cognitive decline, MCI, and presymptomatic FTD). Accordingly, these structure–function signatures should be interpreted as markers of dementia-stage burden. Future longitudinal cohorts uniformly spanning the disease continuum and stratifying by AD and FTD subtypes are needed to assess early-stage sensitivity and subtype-specific structure-function-cognition profiles. We did not consider staggered relationships between biomarker measures[67]. We did not include patients with mild cognitive impairment or presymptomatic disease and did not assess potential biphasic relationships between atrophy and FC[68,69]. We also did not attempt to identify structure-function components related to typical aging[70,71]. Finally, this study only considered task-free fMRI, leaving open how task-engaged brain activity is disrupted by these atrophy patterns.

Function component 1 (F1) correlated with clinical impairment equally well in FTD and AD. F1 has several desirable biomarker properties[72] including: (1) surrogacy with CDR, the standard primary endpoint in dementia clinical trials[73]; 2) prediction of more rapid longitudinal MMSE decline; (3) longitudinal within-subject correlation with clinical worsening and progressive atrophy; and (4) reproducible relationships with brain structure and clinical severity scores across 37 different sites combining the UCSF and ADNI datasets. In a clinical setting, F1 could be used in tandem with other biomarkers to predict which patients are at greater risk of subsequent cognitive decline. This will require further validation in individuals with subjective cognitive complaints or mild cognitive impairment. In a clinical trial setting, while molecular and anatomical neuroimaging biomarkers are more widely applied[74], fMRI biomarkers have the potential to measure cognition-supporting brain activity with high anatomical precision prior to widespread neurodegeneration. In this study, structural and functional components explained independent variance in CDR. This suggests patients might benefit from treatments that slow neurodegeneration, restore function, or both. Thus, monitoring both structural and functional biomarkers could add value to clinical trials. While the current observational study in a diverse cohort was well-suited for biomarker discovery, a key next step is analytical and clinical biomarker validation. This effort should focus on specified contexts of use, including as a prognostic biomarker to identify individuals at risk for cognitive decline, or as a monitoring biomarker for measuring treatment response[75]. This will ideally include an fMRI acquisition protocol that optimizes within-subject reliability[76,77]. We did not stratify by medication in this study, where participants received routine symptomatic treatments outside clinical trials. However, it will be prudent for future studies to consider the effect of symptomatic therapies on activity imbalance, given that activity gradients reflect neurotransmitter receptor distributions[29,30,78] and may be modulated by acetylcholinesterase inhibitors or selective serotonin reuptake inhibitors commonly used in dementia treatment.

## Methods

### Subject selection

Patients with dementia and cognitively normal control subjects were recruited through ongoing studies at the University of California, San Francisco (UCSF) Memory and Aging Center. All subjects or their surrogates provided informed consent according to the Declaration of Helsinki and the procedures were approved by the UCSF Institutional

Review Board. The study cohort was assembled without exclusion based on race, ethnicity, sex, gender, or socioeconomic status. All subjects underwent a clinical history, physical examination, neuroimaging, and neuropsychological assessment within 90 days of scanning. Cognitively normal control subjects were recruited from the Hillblom Healthy Aging Study with ages between 45-85, minor or no memory problems, a clinical dementia rating score of 0, and no diagnosis of a neurodegenerative disease or other major health condition. Subjects were excluded if they had a significant history of other neurological diseases or structural brain abnormalities inconsistent with their primary clinical syndrome. Subjects were included whether or not they took medication for symptoms of Alzheimer's disease or frontotemporal dementia.

A large initial control dataset ($n = 568$) was subsequently divided for multiple purposes. The initial patient group consisted of patients ($n = 309$) who received a primary clinical diagnosis of either Alzheimer's disease[79], behavioral variant frontotemporal dementia[80], semantic variant or nonfluent variant primary progressive aphasia[81], or corticobasal syndrome[82]. All diagnoses were made within 90 days of the patients' MRI scan. We assigned each patient to a high, intermediate, or low confidence diagnosis group. High confidence subjects had a single clinical diagnosis of one of the five syndromes of interest at one or more clinical visits. Intermediate confidence subjects had a best estimate clinical diagnosis of the syndrome of interest, but additionally either: 1) an alternative possible diagnosis including AD, bvFTD, CBS, nfvPPA, svPPA, progressive supranuclear palsy (PSP), amyotrophic lateral sclerosis (ALS), posterior cortical atrophy (PCA), or logopenic variant PPA (lvPPA) or 2) a best estimate clinical diagnosis that was stable for multiple visits before shifting away from the syndrome of interest in a later visit. These subjects were assigned to the intermediate confidence group if they had three or more clinic visits with a stable primary diagnosis of the syndrome of interest. Low confidence subjects had multiple diagnoses as best estimates, including the syndrome of interest, or a diagnosis for the syndrome of interest that shifted away at the next visit. Our primary analysis focused on 221 high and intermediate confidence patients with dementia, excluding low confidence patients and MRI quality control failures (see below). 100 cognitively normal subjects were selected who passed image quality control and were matched to the overall patient group for age, sex, MRI scanner distribution, and fMRI head motion. The number of subjects with each diagnosis are shown in Table 1. The self-reported race/ethnicity for the 321 subjects included 25 Asian, 5 Black, 5 Hispanic, 263 White, and 23 unreported.

## Neuroimaging acquisition

All subjects were scanned at the UCSF Neuroscience Imaging Center, on either Siemens Trio or Siemens Prisma Fit 3 T MRI scanners. Subjects were scanned one to eight times over the course of their clinic visits. Our main analysis focused on the baseline scan for each subject. The number of patients with each diagnosis scanned on either the Trio or Prisma are shown in Table 1. Subjects received T1-weighted magnetization-prepared rapid gradient echo structural MRI (MPRAGE) scans with similar acquisition parameters on the Trio or Prisma: acquisition time: 8:53; sagittal slice orientation; thickness: 1.0 mm; field of view: 160 x 240 x 256 mm; isotropic voxel resolution: 1 mm³; TR: 2300 ms; TE: 2.98 ms for Trio, 2.9 ms for Prisma; TI: 900 ms, flip angle: 9°.

Task-free fMRI scans were run using a T2*-weighted echoplanar scan with subjects instructed to remain awake with their eyes closed. The parameters on the Trio were: acquisition time: 8:06; axial orientation with interleaved ordering; field of view: 230 x 230x129 mm; matrix size: 92 × 92, effective voxel resolution: 2.5 × 2.5 ×3.0 mm; TR: 2000 ms, for a total of 240 volumes; TE: 27 ms. For the Prisma, the fMRI parameters were: acquisition time: 8:05; axial orientation with interleaved multi-slice mode and multiband acceleration=6; field of

view: 211x211x145 mm; matrix size: 92×92, effective voxel resolution: 2.2 × 2.2 × 2.2 mm; TR: 850 ms, for a total of 560 volumes; TE: 33 ms.

## Structural image processing

MPRAGE scans that passed visual inspection were processed using a structural processing pipeline that has been previously described[83] and is included in the **Supplementary Methods**. Briefly, we used unified normalization/segmentation in SPM12 (version r7771)[84] to obtain gray matter tissue probability maps in MNI152NLin6Asym standard space. Normalized gray matter maps were smoothed with an 8 mm FWHM Gaussian kernel. Voxelwise gray matter atrophy was calculated using a normative model from an independent set of cognitively normal control subjects ($n = 397$). The W-score was the difference between actual gray matter volume and the estimated gray matter volume, divided by the standard deviation of the normative sample. W-scores were averaged across voxels in each of the 246 cortical and subcortical brain regions (see below) and used as the measurement of gray matter atrophy throughout the study.

## Functional image processing

Functional MRI scans were processed using fMRIPrep (version 21.0.0) (Esteban et al., 2019)[85] (RRID:SCR_016216). The full details of the processing pipeline are described in the **Supplementary Methods**. Briefly, the confound timeseries for head motion estimates, CSF, and WM were expanded to include the temporal derivatives and quadratic terms[86]. Bandpass filtering in the frequency range 0.008-0.08 Hz was performed on the confound timeseries and BOLD images using FSL (version 6.0.5) fslmaths and AFNI (version 21.12.19) 3dBandpass. We did not perform global signal regression and instead assessed global signal variance as a disease-relevant variable of interest. The global signal was computed as the mean BOLD signal across the 246 regions (see below) at each time point. Subjects with greater than 0.55 mm mean FD were excluded from subsequent analysis[87]. From the pool of all fMRI scans for all available control and patient subjects ($n = 1591$), this resulted in the exclusion of 194 scans (12.2%). Scans were also screened by applying data-driven PCA-based outlier detection to FC matrices[88] and excluding scans ( > 1 standard deviation above the mean on the first principal component). This excluded 288 scans and left an available pool of 1108 scans.

## Scanner harmonization

Structural and functional MRI data from the two different MRI scanners were harmonized using ComBat[89]. ComBat is a batch effect correction method that removes site-specific variance using a location and scale adjustment model[90]. Information from all features is pooled to estimate the statistical properties of batch effects and an empirical Bayes framework is used to improve parameter estimates. This approach has previously proven effective for removing site-specific variance from multi-site structural MRI atrophy data[91] and fMRI functional connectivity data[92] while preserving biologically relevant variance related to disease status, age, and sex. Gray matter mean W-score values were estimated for 246 regions of interest (210 cortical, 36 subcortical including the caudate, putamen, globus pallidus, and thalamus) from the Brainnetome atlas[93]. A design matrix was constructed, including MRI scanner as the main batch variable and covariates for patient/control status, age, and sex. ComBat was then used to harmonize the W-score values for each region. For functional MRI, mean BOLD timeseries for each scan were obtained for each region and entered into a temporal PCA (see next section). Using the 246 temporal component timeseries, we computed the covariance matrix. The upper triangle from each [246 ×246] covariance matrix was extracted, including the diagonal, flattened into a [30381 ×1] vector, and stacked for all subjects. The [321 ×30381] matrix was run through ComBat with the same design matrix. The harmonized covariance values were then used to derive harmonized FC matrices.

### Gradient derivation and functional connectivity analysis

We studied the low-dimensional basis of functional connectivity patterns by performing dimensionality reduction on fMRI BOLD timeseries data. Specifically, we derived activity gradient spatial maps and temporal activity timeseries[30]. Here, we obtained an independent cohort of cognitively normal subjects ($n = 321$) that were age, sex, and scanner-matched to our primary cohort of 221 patients and 100 cognitively normal subjects. The 246 regional mean BOLD timeseries for the independent cohort subjects were temporally concatenated into a [122795 ×246] matrix, and PCA was performed. This yielded a brain activity latent space in which brain region component loadings (eigenvectors scaled by their corresponding eigenvalues) represented each region's weight on that component. We refer to the spatial maps for each component as gradients and the component scores as the gradient timeseries. Our main analysis focused on the first six dimensions with additional consideration of 12 dimensions. The fMRI ROI timeseries matrix for the primary cohort ([119915 ×246]) was projected into this latent space to obtain the gradient timeseries. For each subject, we computed the [246 ×246] gradient covariance matrix. We used these matrices to derive functional connectivity matrices by obtaining each region's variance (summing on-diagonal values across all components), each region pair's covariance (summing off-diagonal values across all component pairs), and using these quantities to compute the region pairwise Pearson correlation coefficients. An independent PCA was run on the primary cohort [119915 ×246] ROI timeseries matrix to validate the spatial gradient pattern reliability.

### Brain structure-function statistical analysis

The relationship between atrophy and brain-wide functional connectivity was assessed in two stages. In the first stage, we vectorized each subject's FC matrix upper triangle into a [30135 ×1] vector and stacked these to obtain the ([321 subject x 30135 edge] group FC matrix. We performed partial least squares regression on the brain atrophy W-scores ([321 subjects x 246 regions]) and the functional connectivity ([321 subject x 30135 edges]). PLSR is asymmetric, designed to decompose the 'X' variable into components that maximally covary with 'Y' variable[94]. For this reason, PLSR 'X' components can be very similar to performing PCA on 'X' data alone. We chose to use atrophy as the 'X', i.e. the "grounding" variable, based on two factors: 1) our stronger hypothesis about the spatial patterns of the atrophy components, and 2) lower variability in structural MRI than functional MRI. The details of the PLSR component reliability and replication across dataset folds and cross-decomposition algorithms are described in the **Supplementary Methods**. Brain images were created using Surf Ice (version 1.0.20211006), Connectome Workbench (version 1.5), and FSLeyes (version 1.0.5).

We visualized structure-function relationships for each component as matrices of PLSR FC edge weights. The 246 regions were grouped into 14 previously defined functional connectivity modules[83], based on a modular partitioning of a group-averaged task-free functional connectivity matrix from 75 healthy older control subjects. These modules included in this partition are: visual, sensory-motor, dorsal attention network, auditory, hippocampal, parahippocampal, amygdala, cingulo-opercular, salience, anterior temporal, left fronto-parietal, default mode network, right fronto-parietal, and subcortical. We created spatial maps summarizing the most prominent FC patterns for each component by computing region-wise sums of the PLSR FC edge weight matrices.

Syndrome-associated atrophy and functional connectivity patterns were assessed by identifying the subset of patients expressing the typical pattern. We determined this by performing linear discriminant analysis on structure component 1-3 scores with syndrome label as the response variable. This resulted in 51/82 AD patients, 25/41 bvFTD, 10/27 CBS, 9/34 nfvPPA, 32/37 svPPA, and 76/100 CN. The mean atrophy map and FC matrix were computed for each group, and (syndrome – cognitively normal) alterations were derived. The reconstructed FC matrix for each group was computed as the outer product of function component 1-3 scores and the corresponding loadings. Correlations were measured between the typical mean actual FC matrix for each syndrome and the reconstructed FC matrix.

Statistical correction for multiple comparisons was performed using FDR correction for the number of tests in a given analysis, using $q = 0.05$. Statistical significance was reported based on FDR $p < 0.05$ unless otherwise specified. The number of tests performed was: structure-function correlations, $n = 3$; true FC versus reconstructed FC matrix, $n = 5$; functional connectivity variance versus gradient variance, $n = 21$; ADNI models: $n = 7$; true versus simulated FC based on coupling parameter eigenmodes: $n = 1$ global test; eigenmode gradient amplitude/angle and gradient variance/covariance: $n = 1$ global test and 21 specific tests; spatial gradients in discovery and replication datasets: $n = 78$.

### Dynamical systems modeling

We used dynamical systems modeling to analyze gradient dynamic activity in different groups of subjects[30]. All these analyses used the first six gradients. For each gradient timeseries, the first and second derivatives were calculated by finite differencing using the 'gradient' function in MATLAB (version 2022b). We then ran linear regression for each gradient to estimate its second derivative timeseries ($G''$) as a function of all six gradients' timeseries ($G$) and first derivatives ($G'$) along with an intercept. The parameter estimates (coupling parameters) for the 13 terms from each regression were then used to define a system of six coupled second-order ordinary differential equations:

$$G1'' = \beta_{1,0} + \beta_{G1,1}G1 + \beta_{G1',1}G1' + \beta_{G2,1}G2 + \beta_{G2',1}G2' + \cdots$$
$$+ \beta_{G6,1}G6 + \beta_{G6',1}G6'$$

$$G2'' = \beta_{2,0} + \beta_{G1,2}G1 + \beta_{G1',2}G1' + \beta_{G2,2}G2 + \beta_{G2',2}G2' + \cdots$$
$$+ \beta_{G6,2}G6 + \beta_{G6',2}G6' \ldots$$

$$G6'' = \beta_{6,0} + \beta_{G1,6}G1 + \beta_{G1',6}G1' + \beta_{G2,6}G2 + \beta_{G2',6}G2' + \cdots$$
$$+ \beta_{G6,6}G6 + \beta_{G6',6}G6'$$

These equations modeled gradient interactions as a system of linear coupled harmonic oscillators with damping[95].

Eigendecomposition was used to analyze the governing dynamics of the harmonic oscillator system. We transformed each second-order equation to a set of two first-order equations using substitution. This linear system can be represented as:

$$\frac{d\mathrm{Y}}{dt} = A\mathrm{Y}$$

where Y is the gradient timeseries and A is the coupling parameter matrix. Eigendecomposition of the coupling parameter matrix results in:

$$A\mathrm{V} = \lambda\mathrm{V}$$

where $\lambda$ and V are the eigenvalues and eigenvectors for the $m$ eigenmodes. Each eigenmode describes a signal oscillating at a specific frequency. Eigendecomposition of a harmonic oscillator system with damping typically yields complex eigenvalues, with the real and imaginary parts ($\alpha + i\beta$) describing the damping (exponential growth or decay) and frequency of each eigenmode, respectively. The solution to

the differential equation with complex eigenvalues is:

$$Y(t) = e^{\lambda t} V$$

$$Y(t) = e^{(\alpha + i\beta)t} V$$

$$Y(t) = (e^{\alpha t} \cos \beta t + i e^{\alpha t} \sin \beta t) V$$

Positive or negative values of α represent the damping of each eigenmode over time. β is the angular frequency (number of cycles per time unit) and is converted to hertz by β/2π/TR. The eigenvector components are also complex, with the real and imaginary parts describing the amplitude and phase angle of each gradient on that eigenmode. When solving the equation for a given set of initial conditions, the real and imaginary parts of each eigenmode are each scaled by constants $k$ to satisfy the initial conditions, e.g.:

$$Y(t) = k_1 \begin{pmatrix} -e^{\alpha t} \sin \beta t \\ e^{\alpha t} \cos \beta t \end{pmatrix} + k_2 \begin{pmatrix} e^{\alpha t} \cos \beta t \\ e^{\alpha t} \sin \beta t \end{pmatrix}$$

For a given gradient $G$, the overall solution for its timeseries on an eigenmode is:

$$Y_G(t) = k_1 \begin{pmatrix} -a e^{\alpha t} \sin \beta t \\ b e^{\alpha t} \cos \beta t \end{pmatrix} + k_2 \begin{pmatrix} a e^{\alpha t} \cos \beta t \\ b e^{\alpha t} \sin \beta t \end{pmatrix}$$

where a and b are the real and imaginary parts of the gradient's eigenvector component on eigenmode $m$.

The coupling parameter matrix was computed for each subject, from which eigenmodes were derived and gradient time series were simulated. Details on the gradient timeseries simulations are described in the **Supplementary Methods**, including for the subjects with the lowest and highest overall atrophy.

The eigenmodes for each subject were used to measure two across-eigenmode quantities: (1) each gradient's cumulative amplitude (2) each gradient pair's cumulative phase angle. A gradient's cumulative amplitude was computed as:

$$\sum_{m=1}^{n} \sqrt{(-a+b)^2 + (a+b)^2}$$

The phase angle difference between a pair of gradients on a given mode was calculated by subtracting the angles for $a_1 + i b_1$ and $a_2 + i b_2$. The cumulative phase angle difference between a pair of gradients across eigenmodes was then measured as the circular average of angle differences using the 'circ_mean' function in the 'circstat' MATLAB package[96], weighted by the product of the gradient amplitudes on the respective eigenmodes. Gradient amplitudes and gradient-pair angles (21 measurements per subject) were then correlated across subjects with gradient variance and covariances (21 measurements per subject), resulting in a [21 × 21] correlation matrix. Eigenvector angle and amplitude plots were generated using MATLAB.

## Neuropsychological testing
Each subject completed a neuropsychological battery. This included assessments for global cognition and function with the Clinical Dementia Rating CDR® + NACC-FTLD[36] and the Mini-Mental Status exam (MMSE)[97]. The battery also tested the following domains: episodic memory with the short form California Verbal Learning Test (CVLT)[98] and the Benson figure delayed recall from the uniform data set[99]; working memory with the forward and backward digit span length tests; processing speed with the trail making test (Trails A) and Stroop naming tests[100]; executive function with the Stroop

interference test, Delis–Kaplan Executive Function System (DKEFS) Design Fluency[101], and the modified trail making test (Trails B); visuospatial processing with the Benson figure copy test[102] and Number Location subtest of the Visual Object and Space Perception battery (VOSP)[103]; and speech and language with the Boston Naming Test (BNT)[104], the Peabody Picture Vocabulary Test (PPVT)[105], the animal and vegetable naming tests, the D-letter naming test, syntax comprehension[106], and verbal articulatory agility[107].

## Brain-behavior statistical analysis
Brain-behavior relationships were estimated with generalized additive models using the 'mgcv' package[108] in R. We selected only tests where 120 or more subjects had scores. For each test, we set up a model with predictors of atrophy scores for the first three atrophy components and functional scores associated with the first three atrophy components (from ridge regression). Each of these terms was restricted to have a maximal non-linear basis of 3 by setting k = 3 in the model formula. We did not include explicit structure-by-function interaction terms because non-linear terms estimated by a GAM implicitly capture differences in slope across the range of each variable. Each model included linear covariates for age, sex, years of education, scanner type (Trio vs Prisma), and fMRI head motion (mean framewise displacement). Models were fit for 24 cognitive tests, and predictors were deemed significant when their F-statistic-associated $p$-values survived global FDR correction (24 tests x 6 brain-based predictors = 148 terms) with q = 0.05.

The longitudinal UCSF cognitive dataset included 269 individuals (75 cognitively normal, 194 patients) and 359 total timepoints (25 cognitively normal with longitudinal data, 65 patients with longitudinal data). We used a linear mixed effects model to predict MMSE score as a function of S1, F1, time in days, the two and three-way interactions of these terms, along with age, sex, scanner, years of education, control/patient status, and random intercepts for each subject.

Details on the brain-behavior statistical models for the ADNI validation dataset are described in the **Supplementary Methods**.

## Replication analysis
A replication dataset was compiled from subjects in the ADNI3 study[109], obtained from the Alzheimer's Disease Neuroimaging Initiative (ADNI) database (https://adni.loni.usc.edu/). The ADNI was launched in 2003 as a public-private partnership, led by Principal Investigator Michael W. Weiner, MD. The primary goal of ADNI has been to test whether serial magnetic resonance imaging (MRI), positron emission tomography (PET), other biological markers, and clinical and neuropsychological assessment can be combined to measure the progression of mild cognitive impairment (MCI) and early Alzheimer's disease (AD). For up-to-date information, see www.adni-info.org. We included subjects with a diagnosis of cognitively normal or Alzheimer's disease dementia who received structural MRI and resting state functional MRI scans. A total of 966 visits from 573 subjects (CN n = 500, AD n = 73) at 68 sites were obtained. We included the subset of scans that were from sites with 10 or more subjects, and that satisfied fMRI motion criteria (mean framewise displacement <= 0.55 mm), resulting in 821 scans. These subjects had the following characteristics: diagnosis, CN n = 421, AD n = 56; mean age, CN = 72.9 ± 8.1 years, AD = 75.6 ± 8.1 years; 277 female/200 male; 437 right-handed/40 left-handed; mean interscan interval = 1.75 ± 0.7 years). Further details on the different scanners, scan processing, harmonization, and specific brain-behavior statistical models are described in the **Supplementary Methods**.

## Reporting summary
Further information on research design is available in the Nature Portfolio Reporting Summary linked to this article.

## Data availability

Source data are provided with this paper. All data supporting the figures and tables in this study are available as: 1) a MATLAB file named adftd_structure_function_data.mat containing all necessary data variables on Zenodo at https://doi.org/10.5281/zenodo.16783268 with a legend at https://github.com/jbrown81/structure_function/blob/main/adftd_structure_function.m, 2) Text (.txt) files for neuropsychological data at https://github.com/jbrown81/structure_function/tree/main/cognition_data, or 3) NIfTI (.nii) files for activity gradient maps at https://github.com/jbrown81/structure_function/tree/main/gradient_maps. ADNI data are available to qualified investigators at https://adni.loni.usc.edu/.

## Code availability

All custom code used in this study is available at https://github.com/jbrown81/structure_function under the Apache License 2.0. The code DOI is https://doi.org/10.5281/zenodo.17195434.

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

## Acknowledgements

This work was supported by NIH grants K01AG055698 (J.A.B.), P50AG023501, P01AG019724, the Tau Consortium, the Bluefield Project to Cure FTD, and the Larry L. Hillblom Network Grant for the Prevention of Age-Associated Cognitive Decline (2014-A-004-NET). Data used in the preparation of this article were obtained from the Alzheimer's Disease Neuroimaging Initiative (ADNI) database (adni.loni.usc.edu). We thank the patients, their families, and the volunteers whose help and participation made this work possible.

## Author contributions

J.A.B., A.J.L., K.F., and T.P. performed the research. J.A.B., A.J.L., and W.W.S. wrote the manuscript. J.A.B., A.J.L., K.F., T.P., and A.B.W. processed the data. L.P., A.M.S., and M.L.M. contributed to additional data preparation. S.E.L., A.L.B., K.P.R., G.D.R., M.L.G.T., H.J.R., J.H.K., B.L.M., W.W.S., and the Alzheimer's Disease Neuroimaging Initiative provided resources, funding, and overall support. J.A.B. and W.W.S. conceived and supervised the study. ADNI investigators contributed data only and were not involved in analysis, interpretation, drafting, or approval of the manuscript.

## Competing interests

J.A.B. is a founder of Radiata Inc. and holds a financial interest in the company, which may benefit from the commercialization of biomarker software products. J.A.B. and A.J.L. are the inventors of patent WO2024259443A1 based on this work. The remaining authors declare no competing interests.
