## [Transparent Peer Review file · Nature Communications]

Functional network collapse in neurodegenerative disease

Corresponding Author: Dr Jesse Brown

Version 0:

Reviewer comments:

Reviewer #1

(Remarks to the Author)

The authors tackle a critical question in the field of neurodegeneration - the link between neuropathologies that cause altered structure, changes in brain functional activity, and subsequent clinical symptoms. This work builds on previous seminal publications by this group establishing the link between functional brain networks and patterns of atrophy. I thoroughly enjoyed reading this manuscript, and the study takes an important step forward by linking structure/function associations with clinical symptoms and syndromes.

A key question is how definitive these findings are. The replication cohort is reassuring, although it does not include all diagnostic groups. The analysis critically depends on the PLS analysis to identify structure/function components. I would be reassured if the same components were identified with a different method, e.g. canonical correlation analysis, or variational autoencoders.

In the introduction I would take issue with the phrase "the possibility that different atrophy patterns may disrupt distinct gradients". The atrophy reflects a variety of underlying neuropathologies, so strictly speaking it is not the atrophy itself but the associated neuropathologies disrupting the functional gradients. I would suggest "the possibility that the distribution of neuropathology reflected in atrophy patterns may disrupt distinct gradients".

The diagnostic groups represent a broad selection of neurodegenerative disorders, and the transdiagnostic approach is a strength of this study. The associations of syndromes with atrophy/function components are plausible, and these are replicated using an independent dataset and alternative statistical approach, which is reassuring.

The structure/function components correlate with clinical scores using an appropriate transdiagnostic measure (CDR+NACC-FTLD).

I really like the use of functional gradients, identifying that atrophy disrupted relationships between gradients, providing an important explanation for increased and decreased connectivity in disease. The extension to simulate these gradients to investigate the effect of low and high atrophy is interesting, and the possibility that eigenmodes and gradient angles explain hyper/hypoconnectivity is particularly insightful and novel.

I have no issues with the methods. The neuroimaging acquisition and preprocessing methods applied are standard in the field. The TR of 2 seconds for fMRI looks slightly out of date, but in clinical populations this is not unreasonable given these data will have been acquired over many years. An exclusion rate of 12.2% based on motion artefacts would be expected. It would be helpful to know if those excluded were more severely affected than others, or more likely to come from any specific diagnostic group.

The literature review is comprehensive.

(Remarks on code availability)

Unfortunately the git repository only includes a README file. I therefore cannot review the code, but would be happy to do so.

Reviewer #2

(Remarks to the Author)

The article by Brown and colleagues sought to better understand how signatures of neurodegeneration relate to altered functional connectivity, and how low-dimensional representations describing structure-function relationships can be linked to cognitive outcomes across dementia subtypes. Via application of dynamical systems modelling, the study offers novel mechanistic insights into functional network alterations grounded by subtype-specific atrophy patterns. Overall, this manuscript is well-written and technically robust. The authors should be commended for their attention to detail, clear figures, and inclusion of validation and replication cohorts.

- Overall, the article [understandably] is fairly dense. The authors may want to consider moving some information to the supplement to improve readability/accessibility. As one example, while critical, the validation and replication methods and results could be briefly highlighted and referenced for greater detail in the supplement, providing more focus on the key and novel findings from this work (e.g., temporal phase and amplitude collapse).
- Participants “were included whether or not they took medication for symptoms of Alzheimer’s disease or frontotemporal dementia”. This may not be informative or possible to explore the given relatively small subgroup sample sizes but might be at least worth mentioning in the discussion. It is also unclear if this or what other factors may be contributing to the intermediate and low confidence cases; low confidence cases were excluded, however there is no other mention of how this classification may be impacting the analyses or findings.
- It is a little unclear how “stage” is being defined here. The sample includes dementia subtypes as stated, however, there is only a comparison of these subtypes to control cases and no mention or inclusion of participants in preclinical or prodromal stages. In the discussion (pg.17, p2), it is stated that the first atrophy component captured mean overall mean atrophy, most pronounced in cingulo-insular regions, and served as a proxy for disease stage” – is this more just capturing common atrophy across subtypes as opposed to stage of disease? Is this based on CDR? The adjudication process should also probably be briefly described in the Methods.
- Can the authors clarify the significance of this statement: “97% of regions had significant gray matter atrophy ($W > 1.5$) in at least five subjects, showing that these clinical syndromes collectively involve the entire brain.” Would it also be more useful to state the proportion of participants overall and by group?
- Table 1 – Consider adding key metrics to Table 1 overall and by group (e.g., structure and function scores, W /mean atrophy, mean FC, motion, etc.)
- The authors report that structure scores were correlated with overall mean atrophy ($r=0.994$). What was the correlation between functional scores and mean FC? Additionally, how should the statement that SC components were “nearly identical” to the atrophy-only PCA components be interpreted in relation to the joint decomposition of SC-FC?
- Methods – add statement clarifying use of partial r for ADNI replication analyses.
- The authors state “...atrophy-driven changes in coupling of Gradient 1 with other gradients was mathematically equivalent to specific networks increasing or decreasing their integration with the rest of the brain.” Is this statement suggesting the results of the eigenmode analysis provide a mechanistic summary of measures of network integration/segregation?
- The authors state: “Each functional component could be accounted for by variance in six intrinsic activity gradients, suggesting that the disease-related functional alteration patterns are constrained by the brain’s intrinsic functional architecture.” Is this not expected, somewhat of a circular argument given it is based on intrinsic-FC?
- Consider adding references for the statement (pg.18, p2: “This clustering is consistent with the emerging recognition that diverse brain lesions cause convergent low-dimensional behavioral deficits.”
- The authors state “structure-function components explained 25-50% of the variance in cognitive deficits” – a substantial portion of variance is indeed explained; however, the outcomes are designed specifically to detect the presence/severity of dementia. It is unclear how the current results support the use of fMRI or SC-FC coupling as an early marker of disease before widespread neurodegeneration. Comparisons would need to show temporal progression in participants not yet diagnosed with dementia. It is also uncertain, how much variance these components would explain in the cognitive measures in preclinical/prodromal stages of dementia when there is either subtle or emerging patterns of cognitive impairment. Some discussion of AD-specific subtypes may be warranted.

(Remarks on code availability)

Reviewer #3

(Remarks to the Author)

This paper investigates the relationship between brain atrophy and functional connectivity alterations across the Alzheimer's disease (AD) and frontotemporal dementia (FTD) spectrum using structural and functional MRI data from 221 patients and 100 controls. The authors employ partial least squares regression to identify three principal structure-function components and use eigenmode analysis to explain functional connectivity changes through gradient dynamics. The work demonstrates

technical sophistication and includes valuable replication analyses, but several aspects require clarification and revision. Overall the main strengths are:

- The study includes a well-characterized cohort spanning multiple dementia subtypes with appropriate control matching for age, sex, scanner type, and head motion.
- The eigenmode analysis approach to understanding functional connectivity changes through gradient dynamics represents a sophisticated methodological advancement that provides mechanistic insights into network alterations.
- The authors demonstrate commendable scientific rigor through cross-validation using ridge regression, replication in an independent ADNI dataset (477 subjects across 35 sites) and longitudinal validation showing structure-function relationships track clinical progression
- The functional component F1 shows promising biomarker properties, correlating with clinical severity across different dementia types and demonstrating longitudinal sensitivity to disease progression.
- The figures effectively communicate complex relationships between structure, function, and cognition.

Major Concerns

1. Throughout the manuscript, the authors use causal language that overstates their findings:

Abstract: "Cognitive and behavioral deficits in AD and FTD result from brain atrophy and altered functional connectivity"

Introduction: "Cognitive deficits...FTD result from tissue degeneration in specific brain regions"

This language should be revised to reflect that the studies demonstrate correlational relationships. Brain atrophy and functional connectivity alterations are correlates of cognitive deficits, not necessarily their direct causes. Suggested revision: "Cognitive and behavioral deficits in AD and FTD are associated with brain atrophy and altered functional connectivity." and revise causal language throughout the manuscript to accurately reflect correlational findings

2. The manuscript is highly technical and may be challenging for the broader neuroscience audience to follow.

Specific concerns:

- The concept of "temporal phase and amplitude collapse" in the abstract lacks sufficient explanation. Could be Replaced with the clearer description from the text: "reduced amplitude of specific gradients and collapsed phase angles between gradients"

- Eigenmode analysis methodology could benefit from more intuitive explanation and I suggest adding a brief conceptual explanation of eigenmode analysis in simpler terms.

- The relationship between gradients, eigenmodes, and functional connectivity could be clearer and the authors could consider adding a supplementary methods section with step-by-step explanation of the gradient analysis approach.

3. Statistical Considerations: The large number of statistical tests performed (multiple cognitive measures, brain regions, functional connections) raises concerns about multiple comparisons. While FDR correction is mentioned for neuropsychological analyses, the correction strategy for other analyses could be clearer. I recommend to provide more detailed information about multiple comparison corrections throughout all analyses.

Minor Issues

- The harmonization procedure using ComBat could be explained more clearly, particularly regarding potential effects on the structure-function relationships.

- Ensure consistent use of terminology (e.g., "gradient variance" vs "gradient amplitude") throughout the manuscript.

- Figure quality: While generally excellent, some matrix visualizations could benefit from larger text or clearer legends.

The eigenmode analysis provides valuable mechanistic insights, but the biological interpretation of "phase collapse" could be expanded.

- The potential clinical applications of the F1 biomarker could be discussed more extensively

Overall, this is a technically sophisticated study that makes important contributions to understanding structure-function relationships in neurodegeneration. The methodology is innovative, the analyses are comprehensive, and the replication efforts are exemplary. However, the manuscript would benefit significantly from addressing the causal language issues and improving accessibility for the broader neuroscience community. With these revisions, this work would represent a valuable contribution to the field.

(Remarks on code availability)

There is nothing on github except a readme only with the text structure_function, or at least I could not find anything. Maybe this is an error.

It is critical to verify this before publication.

Version 1:

Reviewer comments:

Reviewer #1

(Remarks to the Author)

The authors have comprehensively and thoughtfully addressed my concerns and those of the other reviewers. I have no further comments.

(Remarks on code availability)

The code is now available in a comprehensive and well documented git repository.

Reviewer #2

(Remarks to the Author)

The authors have addressed all concerns.

(Remarks on code availability)

Code availability and documentation assessed and confirmed (code not run). Instructions appear sufficient.

Reviewer #3

(Remarks to the Author)

I thank the authors for their comprehensive and thoughtful response to all reviewer comments. The revisions substantially address the concerns I raised, particularly the removal of causal language throughout the manuscript, the addition of clearer explanations for eigenmode analysis and gradient concepts, and the improved statistical reporting with proper FDR corrections.

The expanded discussion of study limitations regarding preclinical populations and the addition of longitudinal analyses for the F1 biomarker strengthen the work considerably. The authors have made the manuscript more accessible while maintaining its technical rigor.

I am satisfied with these revisions and believe the paper now makes a valuable contribution to understanding structure-function relationships in neurodegeneration.

(Remarks on code availability)

I confirmed that the authors have now made their code and data publicly available through the provided GitHub repository and Zenodo archive, with well-documented scripts and clear documentation that will facilitate reproducibility and future research in this area.

Manuscript Number.: NCOMMS-25-26312

Title: Functional network collapse in neurodegenerative disease

Corresponding Author: Jesse A. Brown

Authors: Alex J. Lee, Kristen Fernhoff, Taylor Pistone, Lorenzo Pasquini, Amy B. Wise, Adam M. Staffaroni, Maria Luisa Mandelli, Alzheimer's Disease Neuroimaging Initiative, Suzee E. Lee, Adam L. Boxer, Katherine P. Rankin, Gil D. Rabinovici, Maria Luisa Gorno Tempini, Howard J. Rosen, Joel H. Kramer, Bruce L. Miller, William W. Seeley

We thank the editor and the three reviewers for taking the time to review this manuscript. We appreciate your helpful comments and think the paper is improved by these revisions. Please find our responses to each reviewer comment below. For each response where we added or changed text in the manuscript, here we list the page number. We are submitting two copies of the manuscript:

- * revision_1_main_supplementary.pdf: changes shown in blue text
- * revision_1_main_supplementary_clean.pdf: no blue text for changes

The major changes we made are:

- We tested the reproducibility of the structure-function components using an alternative decomposition algorithm, canonical PLS. We were able to reproduce the same three structure and function components.
- We improved manuscript for readability by 1) moving substantial detail about the validation, replication, and neuroimaging processing methods to the Supplementary Results and Methods and 2) adding a conceptual explanation of eigenmodes and a step-by-step guide.
- We clarified our definition of disease stage, which we usually use to refer to atrophy stage. We also extended our assessment of disease stage and subtypes. We demonstrate that the three primary atrophy components capture meaningful stage variability in all syndromes and subtype variability in AD, bvFTD, CBS, and svPPA.
- We removed the causal language throughout and clarified that we are reporting statistical correlations between structure, function, and cognition.
- The code is now available at: https://github.com/jbrown81/structure_function

Reviewer #1 (Remarks to the Author):

The authors tackle a critical question in the field of neurodegeneration - the link between neuropathologies that cause altered structure, changes in brain functional activity, and subsequent clinical symptoms. This work builds on previous seminal publications by this group establishing the link between functional brain networks and patterns of atrophy. I thoroughly enjoyed reading this manuscript, and the study takes an important step forward by linking structure/function associations with clinical symptoms and syndromes.

A key question is how definitive these findings are. The replication cohort is reassuring, although it does not include all diagnostic groups. The analysis critically depends on the PLS

analysis to identify structure/function components. I would be reassured if the same components were identified with a different method, e.g. canonical correlation analysis, or variational autoencoders.

We agree this is an important issue. In response, we considered different algorithms for performing cross-decomposition of atrophy and functional connectivity. We considered CCA but opted against it because it does not perform well when there are more variables than observations, as is the case here with 321 subjects and 30135 variables (functional connectivity edges). We opted for PLSCanonical, an alternative to PLSR which finds common modes shared between X (atrophy) and Y (functional connectivity) in a symmetric, correlation-like way rather than PLSR's asymmetric, regression-like objective of using X to predict Y. We were pleased to find the same set of structure-function components in the same sequence. We describe this new result on Supplementary P. 15:

"We evaluated the reliance of the structure-function components on the cross-decomposition algorithm by comparing our default PLSR method to 'PLSCanonical' in scikit-learn. PLSCanonical finds common modes shared between X (atrophy) and Y (functional connectivity) in a symmetric, correlation-like way rather than PLSR's asymmetric, regression-like objective of using X to predict Y. Nonetheless, we found that the first three structure-function components from PLSCanonical corresponded to those from PLSR, with the X and Y components matching in order (S1: $r=0.70$; S2: $r=0.90$, S3: $r=0.91$; F1: $r=0.99$; F2: $r=0.96$; F3: $r=0.95$). Thus the structure-function components appear robust to the choice of the algorithm."

In the introduction I would take issue with the phrase "the possibility that different atrophy patterns may disrupt distinct gradients". The atrophy reflects a variety of underlying neuropathologies, so strictly speaking it is not the atrophy itself but the associated neuropathologies disrupting the functional gradients. I would suggest "the possibility that the distribution of neuropathology reflected in atrophy patterns may disrupt distinct gradients". Thank you for making this clarifying distinction. On P. 2, we now say: "raising the possibility that specific neuropathology patterns and their resultant atrophy may disrupt distinct gradients".

The diagnostic groups represent a broad selection of neurodegenerative disorders, and the transdiagnostic approach is a strength of this study. The associations of syndromes with atrophy/function components are plausible, and these are replicated using an independent dataset and alternative statistical approach, which is reassuring.

The structure/function components correlate with clinical scores using an appropriate transdiagnostic measure (CDR+NACC-FTLD).

I really like the use of functional gradients, identifying that atrophy disrupted relationships between gradients, providing an important explanation for increased and decreased connectivity in disease. The extension to simulate these gradients to investigate the effect of low and high atrophy is interesting, and the possibility that eigenmodes and gradient angles explain hyper/hypoconnectivity is particularly insightful and novel.

I have no issues with the methods. The neuroimaging acquisition and preprocessing methods applied are standard in the field. The TR of 2 seconds for fMRI looks slightly out of date, but in clinical populations this is not unreasonable given these data will have been acquired over many years. An exclusion rate of 12.2% based on motion artefacts would be expected. It would be helpful to know if those excluded were more severely affected than others, or more likely to come from any specific diagnostic group.

We have added an assessment of the characteristics for the excluded vs. included scans on Supplementary P. 14: “The excluded scans included all diagnoses (33 AD, 31 bvFTD, 18 CBS, 12 nfvPPA, 41 svPPA, 347 CN) and did not differ from the included scans in terms of mean atrophy (excluded mean region W-score=0.29 vs included mean W-score=0.26). There was a slight trend for excluded scans to have a lower CDR score than the included scans (mean=3.16 vs 2.66, $t=1.83$, $p=0.07$).”

Overall, we did not find any major or concerning differences in the clinical or anatomical characteristics of the excluded scans.

The literature review is comprehensive.

Reviewer #1 (Remarks on code availability):

Unfortunately the git repository only includes a README file. I therefore cannot review the code, but would be happy to do so.

We apologize for the missing code. The code is now at:

https://github.com/jbrown81/structure_function

and the data is at:

<https://zenodo.org/records/16783268>

The main two scripts are:

https://github.com/jbrown81/structure_function/blob/main/adftd_structure_function.m

and

https://github.com/jbrown81/structure_function/blob/main/adftd_structure_function_prep.m

Reviewer #2 (Remarks to the Author):

The article by Brown and colleagues sought to better understand how signatures of neurodegeneration relate to altered functional connectivity, and how low-dimensional representations describing structure-function relationships can be linked to cognitive outcomes across dementia subtypes. Via application of dynamical systems modelling, the study offers novel mechanistic insights into functional network alterations grounded by subtype-specific atrophy patterns. Overall, this manuscript is well-written and technically robust. The authors should be commended for their attention to detail, clear figures, and inclusion of validation and replication cohorts.

- Overall, the article [understandably] is fairly dense. The authors may want to consider moving some information to the supplement to improve readability/accessibility. As one example, while critical, the validation and replication methods and results could be briefly highlighted and referenced for greater detail in the supplement, providing more focus on the key and novel findings from this work (e.g., temporal phase and amplitude collapse).

Thank you for this suggestion. We have moved substantial detail about the validation, replication, and neuroimaging processing methods to the Supplementary Results and Methods, leaving the most salient information in the main text. Below are the sections we moved to the supplement, showing the Supplementary page number and section title:

- P. 9/10: *Structure-function component validation*
- P. 10: *Structure-function-cognition reliability*
- P. 11: *Supplementary Results for Supplementary Figure 9.*
- P. 12: *Structural image processing*
- P. 12/13: *Functional image processing*
- P. 13/14: *Brain structure-function statistical analysis*
- P. 14/15: *Dynamical systems modeling*
- P. 15: *Brain-behavior statistical analysis*
- P. 15/16: *Replication analysis*

- Participants “were included whether or not they took medication for symptoms of Alzheimer’s disease or frontotemporal dementia”. This may not be informative or possible to explore the given relatively small subgroup sample sizes but might be at least worth mentioning in the discussion. It is also unclear if this or what other factors may be contributing to the intermediate and low confidence cases; low confidence cases were excluded, however there is no other mention of how this classification may impacting the analyses or findings.

Thank you for the reminder to explain the limitations of our cohort selection approach. We added these statements:

- P. 17: “Indeed, our choice to only include patients with a high or medium confidence clinical diagnosis, while intended to create a clean cohort well-suited for structure-function characterization, may have exaggerated the presence of distinct clinical-anatomical subgroups.”

P. 19: “We did not stratify by medication in this study, where participants received routine symptomatic treatments outside clinical trials, with heterogeneous, non-randomized exposure. However, it will be prudent for future studies to consider the effect of symptomatic therapies”

• It is a little unclear how “stage” is being defined here. The sample includes dementia subtypes as stated, however, there is only a comparison of these subtypes to control cases and no mention or inclusion of participants in preclinical or prodromal stages. In the discussion (pg.17, p2), it is stated that the first atrophy component captured mean overall mean atrophy, most pronounced in cingulo-insular regions, and served as a proxy for disease stage” – is this more just capturing common atrophy across subtypes as opposed to stage of disease? Is this based on CDR? The adjudication process should also probably be briefly described in the Methods. Yes, we thank the reviewer for raising this issue about how “stage” is defined. When we use the term “stage” in the paper, we are usually referring to atrophy severity unless otherwise specified. We’ve updated all uses of the term “stage” to specify whether we are referring to atrophy stage or clinical stage.

The reviewer is correct that the first atrophy component is capturing the mean (common) atrophy pattern across subjects. We removed the statement “and served as a proxy for disease stage”. We now clarify this on P. 17: “The first atrophy component captured mean overall atrophy. This corresponded to a mean atrophy pattern, most pronounced in cingulo-insular regions, consistent with the FTD-predominant composition of this dataset (39).”

We have added a more thorough analysis of stage and subtype. The new results and methods are shown below. We find evidence for the presence of meaningful stages and subtypes. These are the changes we’ve made to the main text:

P. 6: “We also tested for the presence of meaningful atrophy stages and subtypes within each syndrome. We found evidence for significant atrophy stage variation within each syndrome, and for the presence of two likely atrophy subtypes in AD, bvFTD, CBS, and svPPA (Supplementary Results and Supplementary Figure 4).”

P. 17: “These three components collectively stratified patients with different syndromes, atrophy stages, and subtypes, consistent with well-characterized atrophy subtypes in different AD and FTD syndromes (41, 42, 5, 43–47).”

These are the new results and methods:

Supplementary Results, P. 9/10: “Across syndromes, between-subject dispersion in global atrophy exceeded controls (all FDR-corrected $q < 0.05$), indicating greater within-syndrome heterogeneity. Within each syndrome, mean atrophy was strongly predicted by three structural components ($R^2 > 0.8$; FDR-corrected $q < 0.05$), consistent with a continuous severity spectrum. Conditioning on this severity axis, model comparison favored two subtypes over one in AD ($\Delta BIC=68.1$, 21/61 subjects in cluster 1/2), with more widespread cortical versus limbic-focused atrophy; bvFTD ($\Delta BIC=4.0$, 34/7 subjects in cluster 1/2), with more ventral versus dorsal frontal atrophy; CBS ($\Delta BIC=14.7$, 10/17 subjects in cluster 1/2), with more widespread versus focal

sensorimotor atrophy; and svPPA ($\Delta\text{BIC}=39.4$, 33/4 subjects in cluster 1/2), with more left-lateralized versus right-lateralized anterior temporal atrophy; but not nfvPPA ($\Delta\text{BIC}=-4.3$), indicating discrete variation beyond severity in four syndromes.”

Supplementary Methods P.17/18: “We tested for the presence of meaningful atrophy stages and subtypes within each syndrome. We analyzed each syndrome separately (AD, bvFTD, CBS, nfvPPA, svPPA, CN) using the three PLS structure components. We used two complementary measures of disease stage: mean overall atrophy, a more coarse measure of disease stage; and Euclidean distance to the cognitively normal group mean, a more context-specific measure of disease stage. We first tested whether mean overall atrophy was captured by the three structure components. We fit a linear model within each syndrome: $\text{mean_atrophy} \sim S1 + S2 + S3$ (with intercept), and calculated the model R^2 and p-value. We also tested whether mean overall atrophy variability in that syndrome exceeded controls using a one-sided F-test on variances (syndrome vs CN). We considered stages present when the regression was significant with a non-trivial effect ($p < 0.05$ and $R^2 \geq 0.10$) and the variance exceeded controls ($p < 0.05$).

After establishing a relationship between mean overall atrophy and the structure components, we derived a proxy for disease stage by computing each subject’s Euclidean distance from the CN centroid, which we refer to as ‘CN distance’. We used this measure to evaluate the possibility of syndrome-specific atrophy subtypes. We removed stage effects within each syndrome by orthogonalizing S1/S2/S3 against CN distance. With these residuals, we tested whether two clusters fit better than one using two metrics: 1) the mean silhouette for $k=2$ (k-means), and 2) the BIC difference for Gaussian mixtures, $\Delta\text{BIC} = \text{BIC}(1\text{-component}) - \text{BIC}(2\text{-component})$, using `fitgmdist` in Matlab. We called atrophy subtypes present when $\Delta\text{BIC} > 0$ and mean silhouette ≥ 0.25 .”

Regarding preclinical or prodromal stages, please see our comment response below.

- Can the authors clarify the significance of this statement: “97% of regions had significant gray matter atrophy ($W > 1.5$) in at least five subjects, showing that these clinical syndromes collectively involve the entire brain.” Would it also be more useful to state the proportion of participants overall and by group?

Thank you for pointing out that this statement was insufficient to capture the diversity of atrophy patterns in this cohort. We’ve updated the assessment as follows with new text on P. 4 and new Supplementary Figure 1 to support the idea that these clinical syndromes collectively involved most of the brain and were well-suited for examining brain-wide structure-function relationships: P. 4: “We first aimed to demonstrate that these clinical syndromes collectively involved most of the brain and were well-suited for examining brain-wide structure-function relationships. We flagged a region as ‘affected’ within a syndrome when five or more patients in that group showed an atrophy W -score greater than 1.5 (**Supplementary Figure 1**). Each syndrome affected an average of 85 regions, and collectively across the five syndromes 217 out of 246 regions were affected. The mean between-syndrome Jaccard index was 0.28, illustrating significant diversity in the spatial atrophy patterns.”

- Table 1 – Consider adding key metrics to Table 1 overall and by group (e.g., structure and function scores, W/mean atrophy, mean FC, motion, etc.)

We appreciate the suggestion to add this information. We've added all this information to a new Supplementary Table 1:

Supplementary P. 9: "Supplementary Table 1. Mean scores for each syndrome for structural atrophy W-score, Structure components 1-3 (S1/S2/S3), mean functional connectivity (FC), Function components 1-3 (F1/F2/F3), and fMRI framewise displacement (FD)."

- The authors report that structure scores were correlated with overall mean atrophy ($r=0.994$). What was the correlation between functional scores and mean FC? Additionally, how should the statement that SC components were "nearly identical" to the atrophy-only PCA components be interpreted in relation to the joint decomposition of SC-FC?

The correlation between mean FC and F1/F2/F3 scores was $r=0.30/0.85/-0.38$. This strong relationship between mean FC and F2 can be attributed to F2 being predominantly explained by Gradient 1, as shown in Figure 4B. Because Gradient 1 is the strongest proxy for the global signal – a shared correlation between every region – individuals with a higher F2 score have a higher amplitude global signal, more shared signal between all regions, and thus higher mean FC.

Regarding the question about SC components being "nearly identical" to the atrophy-only PCA components: as stated in our response to Reviewer 1, the PLSR algorithm we used has an asymmetric, regression-like objective of using X (atrophy) to predict Y (functional connectivity). It starts with a decomposition of X and iteratively finds joint decompositions with Y. This is in contrast with other algorithms like PLSCanonical that find common modes shared between X (atrophy) and Y (functional connectivity) in a symmetric, correlation-like way. We've added this context on P. 24: "For this reason, PLSR 'X' components can be very similar to performing PCA on 'X' data alone."

- Methods – add statement clarifying use of partial r for ADNI replication analyses.

Thank you for catching this. We mislabeled these as partial r correlation coefficients. They were in fact just standard correlation coefficients. We removed the "partial" labels.

- The authors state "...atrophy-driven changes in coupling of Gradient 1 with other gradients was mathematically equivalent to specific networks increasing or decreasing their integration with the rest of the brain." Is this statement suggesting the results of the eigenmode analysis provide a mechanistic summary of measures of network integration/segregation?

Yes, we agree with this statement. As we mentioned above, because Gradient 1 is a proxy for the global signal, if another gradient like Gradient 2 becomes more coupled Gradient 1 and therefore the global signal, then the regions with positive weights on Gradient 2 will become more positively coupled with the rest of the brain. This is equivalent to more network integration. Conversely, regions with negative weights on Gradient 2 will become more negatively coupled with the rest of the brain, i.e. more network segregation. This relationship, where tighter gradient coupling necessarily entails altered integration and segregation of specific networks, is what we

meant by saying “mathematically equivalent”. Regarding mechanisms, we refer to our statement in the Discussion on P.18: “The linkage between functional connectivity, activity gradients, and eigenmodes raises a question about whether any of these processes are epiphenomenal.”

- The authors state: “Each functional component could be accounted for by variance in six intrinsic activity gradients, suggesting that the disease-related functional alteration patterns are constrained by the brain’s intrinsic functional architecture.” Is this not expected, somewhat of a circular argument given it is based on intrinsic-FC?

We thank the reviewer for this thoughtful question. The following is our explanation for why we don’t think this is a circular argument. The six “intrinsic” activity gradients were derived from a separate cohort of cognitively normal subjects. If the disease-related functional alteration patterns *were not* constrained by the brain’s intrinsic functional architecture, then we would expect those six gradients to explain relatively little variance. Instead, new disease-specific components would emerge. The fact that these six gradients *did* explain most of the variance suggests to us that what occurs in this cohort of patients is a reconfiguration of the existing intrinsic gradients. We’ve modified this statement on P. 17: “Each functional component could be accounted for by variance in six intrinsic activity gradients **derived from cognitively normal individuals**”.

- Consider adding references for the statement (pg.18, p2: “This clustering is consistent with the emerging recognition that diverse brain lesions cause convergent low-dimensional behavioral deficits.”

Thank you for pointing out this omission. We now refer to two papers here on P. 18: “This clustering is consistent with the emerging recognition that diverse brain lesions can converge on low-dimensional behavioral deficits (22, 63).”

22. L. Pini, S. C. de Lange, F. B. Pizzini, I. Boscolo Galazzo, R. Manenti, M. Cotelli, S. Galluzzi, M. S. Cotelli, M. Corbetta, M. P. van den Heuvel, M. Pievani, A low-dimensional cognitive-network space in Alzheimer’s disease and frontotemporal dementia. *Alzheimer’s Research & Therapy* 14, 199 (2022).

63. M. Corbetta, J. S. Siegel, G. L. Shulman, On the low dimensionality of behavioral deficits and alterations of brain network connectivity after focal injury. *Cortex* 107, 229–237 (2018).

- The authors state “structure-function components explained 25-50% of the variance in cognitive deficits” – a substantial portion of variance is indeed explained; however, the outcomes are designed specifically to detect the presence/severity of dementia. It is unclear how the current results support the use of fMRI or SC-FC coupling as an early marker of disease before widespread neurodegeneration. Comparisons would need to show temporal progression in participants not yet diagnosed with dementia. It is also uncertain, how much variance these components would explain in the cognitive measures in preclinical/prodromal stages of dementia when there is either subtle or emerging patterns of cognitive impairment. Some discussion of AD-specific subtypes may be warranted.

We thank the reviewer for raising a key point about the cohort composition. We agree that this cohort limits our ability to know how well these structure-function-cognition patterns extend into

the preclinical/prodromal disease stages. We have added this limitation in the Discussion on P. 19: “Our cohort included individuals with a clinical dementia diagnosis and age-matched cognitively normal controls, excluding preclinical and prodromal stages (subjective cognitive decline, MCI, presymptomatic FTD). Accordingly, these structure–function signatures should be interpreted as markers of dementia-stage burden. Future longitudinal cohorts uniformly spanning the disease continuum and stratifying by AD and FTD subtypes are needed to assess early-stage sensitivity and subtype-specific structure-function-cognition profiles.”

Reviewer #3 (Remarks to the Author):

This paper investigates the relationship between brain atrophy and functional connectivity alterations across the Alzheimer's disease (AD) and frontotemporal dementia (FTD) spectrum using structural and functional MRI data from 221 patients and 100 controls. The authors employ partial least squares regression to identify three principal structure-function components and use eigenmode analysis to explain functional connectivity changes through gradient dynamics. The work demonstrates technical sophistication and includes valuable replication analyses, but several aspects require clarification and revision.

Overall the main strengths are:

- The study includes a well-characterized cohort spanning multiple dementia subtypes with appropriate control matching for age, sex, scanner type, and head motion.
- The eigenmode analysis approach to understanding functional connectivity changes through gradient dynamics represents a sophisticated methodological advancement that provides mechanistic insights into network alterations.
- The authors demonstrate commendable scientific rigor through cross-validation using ridge regression, replication in an independent ADNI dataset (477 subjects across 35 sites) and longitudinal validation showing structure-function relationships track clinical progression
- The functional component F1 shows promising biomarker properties, correlating with clinical severity across different dementia types and demonstrating longitudinal sensitivity to disease progression.
- The figures effectively communicate complex relationships between structure, function, and cognition.

Major Concerns

1. Throughout the manuscript, the authors use causal language that overstates their findings: Abstract: "Cognitive and behavioral deficits in AD and FTD result from brain atrophy and altered functional connectivity"

Introduction: "Cognitive deficits...FTD result from tissue degeneration in specific brain regions"

This language should be revised to reflect that the studies demonstrate correlational relationships. Brain atrophy and functional connectivity alterations are correlates of cognitive deficits, not necessarily their direct causes. Suggested revision: "Cognitive and behavioral deficits in AD and FTD are associated with brain atrophy and altered functional connectivity." and revise causal language throughout the manuscript to accurately reflect correlational findings
Thank you for highlighting this important issue. We have updated the excessively causal language throughout the manuscript, especially replacing instances of "caused", "resulted from", and "driven by".

P. 1: "Eigenmode analysis showed that atrophy relates to dampened gradient amplitudes and narrowed phase angles between gradients"

P. 2: "Cognitive deficits in Alzheimer-type dementia (AD) and frontotemporal dementia (FTD) are associated with tissue degeneration in specific brain regions (1–11)."

P. 2: “raising the possibility that specific neuropathology patterns and their associated atrophy may be linked to distinct gradients disruptions and hypo and hyper-connectivity as two sides of the same coin.”

P. 10: “The second group included dysfunctions in executive function, processing speed, language production, and visuospatial processing, associated most strongly with S1 and S2.”

P. 11: “Thus, atrophy-associated changes in coupling of Gradient 1 with other gradients”

P. 17: “These structural and functional alterations collectively correlated with impairments in global and domain-specific cognition.”

P. 18: “can associate with convergent alterations in these functional systems.”

P. 18: “This suggests that focal structural damage can associate with non-local disruptions of large-scale network dynamics.”

P. 18: “This clustering is consistent with the emerging recognition that diverse brain lesions can converge on low-dimensional behavioral deficits (22, 63).”

P. 18: “Across FTD and AD dementia syndromes, functional connectivity levels have been shown to compensate for atrophy or molecular pathology to preserve cognitive functioning (21, 65, 66).”

P. 18: “and executive function, processing speed, language production, and visuospatial processing, correlated with S1 and S2”

2. The manuscript is highly technical and may be challenging for the broader neuroscience audience to follow.

Specific concerns:

- The concept of “temporal phase and amplitude collapse” in the abstract lacks sufficient explanation. Could be Replaced with the clearer description from the text: “reduced amplitude of specific gradients and collapsed phase angles between gradients”
- Eigenmode analysis methodology could benefit from more intuitive explanation and I suggest adding a brief conceptual explanation of eigenmode analysis in simpler terms.
- The relationship between gradients, eigenmodes, and functional connectivity could be clearer and the authors could consider adding a supplementary methods section with step-by-step explanation of the gradient analysis approach.

We recognize the reviewer’s concern that some of our language was excessively technical.

We’ve addressed this with the following three changes:

P. 1/Abstract: “Eigenmode analysis showed that atrophy relates to dampened gradient amplitudes and narrowed phase angles between gradients, offering a mechanistic account of network collapse in neurodegeneration.”

We added a more intuitive explanation of eigenmode analysis on P. 13: “We modeled the low-dimensional brain activity patterns (“gradients”) as a coupled oscillator system: each gradient behaves like a source that fluctuates over time according to its own dynamics and its interactions with the other sources (Methods). We obtained these sources empirically using data-driven dimensionality reduction (37) and have previously shown that such components correspond to spatial activity gradients (30). From each subject’s gradient timeseries we estimate a linear dynamical system—essentially, how the acceleration of each gradient depends on the current level and velocity of all gradients—which yields a coupling matrix (Figure 5A)

(38). Taking the eigendecomposition of this matrix produces a set of eigenmodes that summarize the system's characteristic patterns of co-fluctuation. Each eigenmode oscillates at a single frequency and assigns every gradient an amplitude (how strongly it participates in that mode) and a phase angle (its timing relative to the others). The observed brain activity at any moment can be expressed as the sum of these modes. Changes in gradient amplitudes or in the phase angles between gradients therefore provide a compact way to describe how disease alters large-scale functional coupling."

We added a step-by-step approach for gradient analysis in the Supplementary Methods P. 18:
“

Gradient-to-Eigenmode Workflow

The following is a summary of the procedure for taking individual subject fMRI data and deriving gradients, eigenmodes, and functional connectivity (FC) metrics.

1. Dimensionality reduction. Concatenate region-level fMRI timeseries from an independent cognitively normal cohort and run PCA; retain first 6 components (spatial “gradients”).
2. Project patient data. Multiply each patient's region timeseries by the gradient loading matrix to obtain 6 gradient timeseries per subject.
3. Build coupled oscillator model. Calculate each gradients' first and second temporal derivatives, the velocity and acceleration. Estimate second order linear ordinary differential equations: each gradient's acceleration (x'') as a linear function of all gradients' positions (x) and velocities (x') (13 parameters \times 6 equations = 6×13 coupling matrix).
4. Eigendecomposition. Compute eigenvectors and eigenvalues of the coupling matrix, which are the eigenmodes; extract gradient specific amplitudes and phase angles. Optionally, simulate gradient timeseries using the coupled oscillator model equations with empirically selected initial conditions for all gradients' positions and velocities.
5. Link to FC. Reconstruct FC matrices from real or simulated gradient timeseries; compare amplitude/phase shifts to empirical hypo/hyper connectivity patterns.

“

3. Statistical Considerations: The large number of statistical tests performed (multiple cognitive measures, brain regions, functional connections) raises concerns about multiple comparisons. While FDR correction is mentioned for neuropsychological analyses, the correction strategy for other analyses could be clearer. I recommend to provide more detailed information about multiple comparison corrections throughout all analyses.

Thank you for raising this key issue and oversight on our part. The main step we have taken is to clarify in the methods that we used FDR correction for all reported p-values and now report the number of statistical tests done for each sub-analysis. Throughout the paper we've now replaced “ $p < 0.001$ ” with “FDR $p < 0.05$ ”.

P. 24: “Statistical correction for multiple comparisons was performed using FDR correction for the number of tests in a given analysis using $q=0.05$. Statistical significance was reported based on FDR $p < 0.05$ unless otherwise specified. The number of tests performed were: structure-function correlations, $n=3$; true FC versus reconstructed FC matrix, $n=5$; functional connectivity

variance versus gradient variance, n=21; ADNI models: n=7; true versus simulated FC based on coupling parameter eigenmodes: n=1 global test; eigenmode gradient amplitude/angle and gradient variance/covariance: n=1 global test and 21 specific tests; spatial gradients in discovery and replication datasets: n=78.”

One relevant change is that for the structure-function-CDR replication model on the ADNI longitudinal data, the p-values are trending and only significant at an uncorrected $p < 0.05$. We now mention this explicitly in the Supplementary Results (P. 11) and have added our rationale for reporting these findings in the Supplementary Methods (P. 16).

P. 10: “Second, we evaluated the longitudinal relationship between S1, F1, and CDR-SB using baseline and follow-up data from a subset of 47 patients and 6 cognitively normal subjects from the main dataset (mean visit interval= 1.1 ± 0.5 years, range=0.4-2.6 years). In this smaller longitudinal model, we found a statistical trend for within-subject CDR-SB change correlating with F1 change ($t=2.05$, $p=0.04$; Supplementary Figure 6) and S1 change ($t=2.15$, $p=0.04$). Between-subject CDR-SB mean had a trend relationship to S1 mean ($F=6.06$, $p=0.02$) and F1 mean ($F=3.34$, $p=0.03$) as expected from the cross-sectional model. A brain-only model showed that overall F1 significantly correlated with S1 mean ($F=21.42$, $p < 0.001$) and had a trending correlation with S1 change ($F=6.21$, $p=0.02$).”

P. 28: “Due to the unique nature of this model as a replication sub-analysis including longitudinal data, we report trend-level statistics with $p < 0.05$ where the direction of the result matched the direction from the larger discovery model.”

Minor Issues

- The harmonization procedure using ComBat could be explained more clearly, particularly regarding potential effects on the structure-function relationships.

We've added the following explanation on P. 23:

“ComBat is a batch effect correction method that removes site-specific variance using a location and scale adjustment model (89). Information from all features is pooled to estimate the statistical properties of batch effects and an empirical Bayes framework is used to improve parameter estimates. This approach has previously proven effective for removing site-specific variance from multi-site structural MRI atrophy data (90) and fMRI functional connectivity data (91) while preserving biologically relevant variance related to disease status, age, and sex.”

- Ensure consistent use of terminology (e.g., "gradient variance" vs "gradient amplitude") throughout the manuscript.

We have addressed this issue in the following ways:

- The term “variance” gets overloaded because of our analysis of both statistical variance, in the sense of variance explained in a model, and temporal variance, in the sense of the standard deviation of the gradient timeseries. In the section Low-dimensional functional

connectivity changes associate with different atrophy components, P. 11-13, we now specify “statistical variance” and “temporal variance” to differentiate the two.

- We removed this confusing sentence: “Henceforth when describing gradient timeseries, we use the terms “variance” and “amplitude” interchangeably, as well as “covariance” and “phase”.”

- We added these two sentences making explicit the relationship between gradient temporal variance and gradient amplitude:

P. 14: “This observation suggested a potential three-way relationship between FC, eigenmodes, and gradient temporal variance/covariance. We hypothesized that two eigenmode-derived quantities would relate to gradient temporal variance/covariance, and thus to atrophy-associated FC alterations: 1) the net amplitude of each gradient across all six eigenmodes, which would be related to gradient temporal variance, and 2) the average phase angle between each pair of gradients across all modes, which would be related to the gradient temporal covariance. We statistically evaluated this by measuring the correlation between eigenmode gradient amplitude/angle and gradient temporal variance/covariance.”

P. 15: “Overall, eigenmode analysis revealed that each atrophy-associated hypo/hyperconnectivity pattern was linked to specific gradient amplitude alterations, reflected in gradient temporal variance changes, and phase angle alterations, reflected in gradient pair temporal covariance changes.”

- Figure quality: While generally excellent, some matrix visualizations could benefit from larger text or clearer legends.

Thank you for this comment. We’ve made the following changes:

- Figure 4: removed the small module labels for Figure 4B matrix insets; added the ‘PLSR weight’ label and colorbar; added the ‘r’ label for Figure 4C-F matrices
- Figure 5: increased the font size for Figure 5A column labels; added the ‘r’ label for Figure 5C matrices

- The eigenmode analysis provides valuable mechanistic insights, but the biological interpretation of “phase collapse” could be expanded.

Thank you for pointing this out. We’ve expanded this statement on P. 18: “In the healthy brain, the optimal set point for each gradient may be temporal orthogonality, maximizing the spatiotemporal segregation of different networks (57) and the potential entropy of the brain (58, 59). Previous studies have shown that a loss of high-frequency local interactions between brain regions can result in phase collapse, hypersynchrony, and rsfMRI overconnectivity (60). Here, one possibility is that structural damage may preferentially impair local functional interactions, leading to phase collapse between gradients and resulting in hypo and hyper-connectivity between different networks as two sides of the same coin.”

- The potential clinical applications of the F1 biomarker could be discussed more extensively. We appreciate the reviewer suggesting this. We added one new statistical model in the paper to test the clinical utility of the F1 biomarker. Please see the description of that model and the results below.

Supplementary P. 11: “First, we tested if baseline S1 and F1 scores could predict future cognitive change in subjects with longitudinal cognitive data. The significant predictors of longitudinal MMSE change over three years were baseline F1 ($t=-1.98$, uncorrected $p= 0.04$; higher baseline F1: trend for lower MMSE), time x baseline F1 ($t=-2.72$, FDR $p < 0.05$; Supplementary Figure 6; higher baseline F1: more rapid MMSE decline), and time x baseline S1 x baseline F1 ($t=-2.06$, uncorrected $p= 0.04$; higher baseline S1 and F1: trend for more rapid MMSE decline). Thus, individuals with worse F1 scores had more rapid cognitive decline.”

P. 19:

- “F1 has several desirable biomarker properties (72) including: ... 2) prediction of more rapid longitudinal MMSE decline
- “In a clinical setting, F1 could be used in tandem with other biomarkers to predict which patients are at greater risk of subsequent cognitive decline. This will require further validation in individuals with subjective cognitive complaints and mild cognitive impairment.”

P. 27: “The longitudinal UCSF cognitive dataset included 269 individuals (75 cognitively normal, 194 patients) and 359 total timepoints (25 cognitively normal with longitudinal data, 65 patients with longitudinal data). We used a linear mixed effects model to predict MMSE score as a function of S1, F1, time in days since baseline, the two and three-way interactions of these terms, along with age, sex, scanner, years of education, control/patient status, and random intercepts for each subject.”

Overall, this is a technically sophisticated study that makes important contributions to understanding structure-function relationships in neurodegeneration. The methodology is innovative, the analyses are comprehensive, and the replication efforts are exemplary. However, the manuscript would benefit significantly from addressing the causal language issues and improving accessibility for the broader neuroscience community. With these revisions, this work would represent a valuable contribution to the field.

Reviewer #3 (Remarks on code availability):

There is nothing on github except a readme only with the text `structure_function`, or at least I could not find anything. Maybe this is an error.

It is critical to verify this before publication.

We apologize for the missing code. The code is now at:

https://github.com/jbrown81/structure_function

and the data is at:

<https://zenodo.org/records/16783268>

The main two scripts are:

https://github.com/jbrown81/structure_function/blob/main/adftd_structure_function.m

and

https://github.com/jbrown81/structure_function/blob/main/adftd_structure_function_prep.m